# Understanding and Addressing the Pitfalls of Bisimulation-based Representations in Offline Reinforcement Learning

**Hongyu Zang[1], Xin Li[1],[*] Leiji Zhang[1], Yang Liu[2], Baigui Sun[2], Riashat Islam[3],**
**Rémi Tachet des Combes[4] , Romain Laroche**
[1] Beijing Institute of Technology, China        [2] Alibaba Group, China
[3] McGill University, Mila, Quebec AI Institute, Canada [4] Wayve, UK
{zanghyu,xinli,ljzhang}@bit.edu.cn
{ly261666,baigui.sbg}@alibaba-inc.com
riashat.islam@mail.mcgill.ca
{remi.tachet,romain.laroche}@gmail.com

## Abstract

While bisimulation-based approaches hold promise for learning robust state representations for Reinforcement Learning (RL) tasks, their efficacy in offline RL tasks has not been up to par. In some instances, their performance has even significantly underperformed alternative methods. We aim to understand why bisimulation methods succeed in online settings, but falter in offline tasks. Our analysis reveals that missing transitions in the dataset are particularly harmful to the bisimulation principle, leading to ineffective estimation. We also shed light on the critical role of reward scaling in bounding the scale of bisimulation measurements and of the value error they induce. Based on these findings, we propose to apply the expectile operator for representation learning to our offline RL setting, which helps to prevent overfitting to incomplete data. Meanwhile, by introducing an appropriate reward scaling strategy, we avoid the risk of feature collapse in representation space. We implement these recommendations on two state-of-the-art bisimulation-based algorithms, MICo and SimSR, and demonstrate performance gains on two benchmark suites: D4RL and Visual D4RL. Codes are provided at https://github.com/zanghyu/Offline_Bisimulation.

## 1  Introduction

Reinforcement learning (RL) algorithms often require a significant amount of data to achieve optimal performance [40, 48, 22]. In scenarios where collecting data is costly or impractical, Offline RL methods offer an attractive alternative by learning effective policies from previously collected data [29, 43, 32, 35, 16, 24]. However, capturing the complex structure of the environment from limited data remains a challenge for Offline RL [4]. This involves pre-training the state representation on offline data and then learning the policy upon the fixed representations [51, 47, 41, 53]. Though driven by various motivations, previous methods can be mainly categorized into two classes: i) implicitly shaping the agent's representation of the environment via prediction and control of some aspects of the environment through auxiliary tasks , *e.g.*, maximizing the diversity of visited states [34, 10], exploring attentive contrastive learning on sub-trajectories [51], or capturing temporal information about the environment [47]; ii) utilizing *behavioral metrics*, such as bisimulation metrics [11, 13, 5], to capture complex structure in the environment by measuring the similarity of behavior on the

---

[*]Correspondence to Xin Li.

37th Conference on Neural Information Processing Systems (NeurIPS 2023).

representations [52, 7]. The former methods have proven their effectiveness theoretically and empirically in Offline settings [41, 47, 51], while the adaptability of the latter approaches in the context of limited datasets remains unclear. This paper tackles this question.

Bisimulation-based approaches, as their name suggests, utilize the bisimulation metrics update operator to construct an auxiliary loss and learn robust state representations. These representations encapsulate the behavioral similarities between states by considering the difference between their rewards and dynamics. While the learned representations possess several desirable properties, such as smoothness [19], visual invariance [54, 1, 52], and task adaptation [56, 37, 46, 8], bisimulation-based objectives in most approaches are required to be coupled with the policy improvement procedure [54, 6, 52]. In Offline RL, pretraining state representations via bisimulation-based methods is supposed to be cast as a special case of on-policy bisimulation metric learning where the behavior policy is fixed so that good performance should ensue. However, multiple recent studies [51, 21] suggest that bisimulation-based algorithms yield significantly poorer results on Offline tasks compared to a variety of (self-)supervised objectives.

In this work, we highlight problems with using the bisimulation principle as an objective in Offline settings. We aim to provide a theoretical understanding of the performance gap in bisimulation-based approaches between online and offline settings:*"why do bisimulation approaches perform well in Online RL tasks but tend to fail in Offline RL ones?"* By establishing a connection between the Bellman and bisimulation operators, we uncover that missing transitions, which often occur in Offline settings, can cause the bisimulation principle to be compromised. This means that the bisimulation estimator can be ineffective in finite datasets. Moreover, we notice that the scale of the reward impacts the upper bounds of both the bisimulation measurement[2] fixed point and the value error. This scaling term, if not properly handled, can potentially lead to representation collapse.

To alleviate the aforementioned issues, we propose to learn state representations based on the expectile operator. With this asymmetric operator predicting expectiles of the representation distribution, we can achieve a balance between the behavior measurement and the greedy assignment of the measurement over the dataset. This results in a form of regularization over the bisimulation measurement, thus preventing overfitting to the incomplete data, and implicitly avoiding out-of-distribution estimation errors. Besides, by considering the specific properties of different bisimulation measurements, we investigate the representation collapse issue for the ones that are instantiated with bounded distances (*e.g.*, cosine distance) and propose a way to scale rewards that reduces collapse. We integrate these improvements mainly on two bisimulation-based baselines, MICo [7] and SimSR [52], and show the effectiveness of the proposed modifications.

The primary contributions of this work are as follows:

- We investigate the potential harm of directly applying the bisimulation principle in Offline settings, prove that the bisimulation estimator can be ineffective in finite datasets, and emphasize the essential role of reward scaling.

- We propose theoretically motivated modifications on two representative bisimulation-based baselines, including an expectile-based operator and a tailored reward scaling strategy. These proposed changes are designed to address the challenges encountered when applying the bisimulation principle in offline settings.

- We demonstrate the superior performance our approach yields through an empirical study on two benchmark suites, D4RL [15] and Visual D4RL [35].

## 2   Related Work

**State representation learning in Offline RL**   Pretraining representations has been recently studied in Offline RL settings, where several studies presented its effectiveness [3, 47, 41, 25]. In this paradigm, we learn state representations on pre-collected datasets before value estimation or policy improvement steps are run. The learned representation can then be used for subsequent policy learning, either online or offline. Some typical auxiliary tasks for pretraining state representations include capturing the dynamical [42] and temporal [47] information of the environment, exploring

---

[2]Since some bisimulation-based approaches do not exactly use metrics but instead of pseudometrics, diffuse metrics or else, we will use the term "measurement" in the following.

attentive contrastive learning on sub-trajectories [51], or improving policy performance by applying data augmentations techniques to the pixel-based inputs [9, 35].

**Bisimulation-based methods**   The pioneer works by [20, 33] aim to overcome the curse of dimensionality by defining equivalence relations between states to reduce system complexity. However, these approaches are impractical as they usually demand an exact match of transition distributions. To address this issue, [12, 14] propose a bisimulation metric to aggregate similar states. This metric quantifies the similarity between two states and serves as a distance measure to allow efficient state aggregation. Unfortunately, it remains computationally expensive as it requires a full enumeration of states. Later, [5] devise an on-policy bisimulation metric for policy evaluation, providing a scalable method for computing state similarity. Building upon this, [54] develop a metric to learn state representations by modeling the latent dynamic transition as Gaussian. [6] further investigate the independent couple sampling strategy to reduce the computational complexity of representation learning, whereas [52] propose to learn state representations built on the cosine distance to alleviate a representation collapse issue. Despite the promising results obtained, one of the major remaining challenges in this paradigm is its dependency on coupling state representation learning with policy training. This is not always suitable for Offline settings, given that obtaining on-policy reward and transition differences is infeasible due to our inability to gather additional agent-environment interactions. To adapt bisimulation-based approaches to Offline settings, one solution is to consider the policy over the dataset as a specific behavior policy, and then apply the bisimulation principle on it to learn state representations in a pretraining stage, thus disentangling policy training from bisimulation-based learning. Notably, although there exist recent studies [51, 42] investigating the potential of bisimulation-based methods to pretrain state representations, it has not yielded satisfactory results yet [51].

## 3   Preliminaries

### 3.1   Offline RL

We consider the standard Markov decision process (MDP) framework, in which the environment is given by a tuple $\mathcal{M} = (\mathcal{S}, \mathcal{A}, T, r, \gamma)$, with state space $\mathcal{S}$, action space $\mathcal{A}$, transition function $T$ that decides the next state $s' \sim T(\cdot|s, a)$, reward function $r(s, a)$ bounded by $[R_{\min}, R_{\max}]$, and a discount factor $\gamma \in [0, 1)$. The agent in state $s \in \mathcal{S}$ selects an action $a \in \mathcal{A}$ according to its policy, mapping states to a probability distribution over actions: $a \sim \pi(\cdot|s)$. We make use of the state value function $V^\pi(s) = \mathbb{E}_{\mathcal{M}, \pi} \left[ \sum_{t=0}^\infty \gamma^t r(s_t, a_t) \mid s_0 = s \right]$ to describe the long term discounted reward of policy $\pi$ starting at state $s$. In the sequel, we use $T_s^a$ and $r_s^a$ to denote $T(\cdot|s, a)$ and $r(s, a)$, respectively. In Offline RL, we are given a fixed dataset of environment interactions that include $N$ transition samples, *i.e.* $\mathcal{D} = \{s_i, a_i, s'_i, r_i\}_{i=1}^N$. We assume that the dataset $\mathcal{D}$ is composed of trajectories generated i.i.d. under the control of a behavior policy $\pi_\beta$, whose state occupancy is denoted by $\mu_\beta(s)$.

### 3.2   Bisimulation-based Update Operator

The concept of bisimulation is used to establish equivalence relations on states. This is done recursively by considering two states as equivalent if they have the same distribution over state transitions and the same immediate reward [30, 20]. Since bisimulation considers worst-case differences between states, it commonly results in "pessimistic" outcomes. To address this limitation, the $\pi$-bisimulation metric was proposed in [5]. This new metric only considers actions induced by a given policy $\pi$ rather than all actions when measuring the behavior distance between states:

**Theorem 1.** *[5] Let $\mathbb{M}$ be the set of all measurements on $\mathcal{S}$. Define $\mathcal{F}^\pi : \mathbb{M} \to \mathbb{M}$ by*

$$\mathcal{F}^\pi(g)(s_i, s_j) = |r_{s_i}^\pi - r_{s_j}^\pi| + \gamma \mathcal{W}(g) \left( T_{s_i}^\pi, T_{s_j}^\pi \right) \tag{1}$$

*where $s_i, s_j \in \mathcal{S}$, $r_{s_i}^\pi = \sum_{a \in \mathcal{A}} \pi(a|s_i) r_{s_i}^a$, $T_{s_i}^\pi = \sum_{a \in \mathcal{A}} \pi(a|s_i) T_{s_i}^a$, and $\mathcal{W}(g)$ is the Wasserstein distance with cost function $g$ between distributions. Then $\mathcal{F}^\pi$ has a least fixed point $g_\sim^\pi$, and $g_\sim^\pi$ is a $\pi$-bisimulation metric.*

Although it is feasible to compute the behavior difference measurement $g_\sim^\pi$ by applying the operator $\mathcal{F}^\pi$ iteratively (which is guaranteed to converge to a fixed point since $\mathcal{F}^\pi$ is a contraction), this approach comes at a high computational complexity due to the Wasserstein distance on the right-hand

side of the equation. To tackle this issue, MICo [6] proposed using an independent couple sampling strategy instead of optimizing the overall coupling of the distributions $T_{s_i}^\pi$ and $T_{s_j}^\pi$, resulting in a novel measurement to evaluate the difference between states. Additionally, SimSR [52] further explored the potentiality of combining the cosine distance with bisimulation-based measurements to learn state representations. Both works can be generalized as:

$$\mathcal{F}^\pi G^\pi(s_i, s_j) = |r_{s_i}^\pi - r_{s_j}^\pi| + \gamma \mathbb{E}_{\substack{s_i' \sim T_{s_i}^\pi \\ s_j' \sim T_{s_j}^\pi}} [G^\pi(s_i', s_j')], \tag{2}$$

and $\mathcal{F}^\pi$ has a least fixed point $G_\sim^\pi$ [3]. The instantiation of $G$ varies in different approaches [6, 52]. For example, in SimSR [52], the cosine distance is used to instantiate $G$ on the embedding space, and the dynamics difference is computed by the cosine distance between the next-state pair $(s_i', s_j')$ sampled from a transition model of the environment. A more detailed description can be found in Appendix C.

**Lemma 2.** *[6] (**Lifted MDP**) The bisimulation-based update operator $\mathcal{F}^\pi$ for $\mathcal{M}$ is the Bellman evaluation operator for a specific lifted MDP.*

Due to this interpretation of the bisimulation-based update operator as the Bellman evaluation operator in a lifted MDP, we can derive certain conclusions about bisimulation by drawing inspiration from policy evaluation methods. In the next section, we will borrow analytical ideas from [17] to prove that the bisimulation-based objective may be ineffective for finite datasets. We summarize all notations in Appendix A and provide all proofs in Appendix D.

# 4 Ineffective Bisimulation Estimators in Finite Datasets

The high-level idea of bisimulation-based state representation learning is to learn state embeddings such that when states are projected onto the embedding space, their behavioral similarity is maintained. We denote our parameterized state encoder by $\phi : \mathcal{S} \to \mathbb{R}^n$ and a distance $D(\cdot, \cdot)$ in the embedding space $\mathbb{R}^n$ by $G_\phi^\pi(s_i, s_j) \doteq D(\phi(s_i), \phi(s_j))$. For instance, $D(\cdot, \cdot)$ may be the Łukaszyk–Karmowski distance [6] or the cosine distance [52]. To avoid unnecessary confusion, we defer implementation details to Section 5.

When considering bisimulation-based state representations, the goal is to acquire stable state representations under policy $\pi$ via the measurement $G_\sim^\pi$. The primary focus is usually to minimize a loss over the *bisimulation error*, denoted by $\Delta_\phi^\pi$, which measures the distance between the approximation $G_\phi^\pi$ and the fixed point $G_\sim^\pi$:

$$\Delta_\phi^\pi(s_i, s_j) := |G_\phi^\pi(s_i, s_j) - G_\sim^\pi(s_i, s_j)|. \tag{3}$$

However, since the fixed point $G_\sim^\pi$ is unobtainable without full knowledge of the underlying MDP, this approximation error is often unknown. Recall that in Lemma 2, we have shown that we can connect a bisimulation-based update operator to a lifted MDP. Taking inspiration from Bellman evaluation for the value function, we define the *bisimulation Bellman residual* $\epsilon_\phi^\pi$ as:

$$\epsilon_\phi^\pi(s_i, s_j) := |G_\phi^\pi(s_i, s_j) - \mathcal{F}^\pi G_\phi^\pi(s_i, s_j)|. \tag{4}$$

Then, we can connect the bisimulation Bellman residual with the bisimulation error by the following:

**Theorem 3.** *(**Bisimulation error upper-bound**). Let $\mu_\pi(s)$ denote the stationary distribution over states, let $\mu_\pi(\cdot, \cdot)$ denote the joint distribution over synchronized pairs of states $(s_i, s_j)$ sampled independently from $\mu_\pi(\cdot)$. For any state pair $(s_i, s_j) \in \mathcal{S} \times \mathcal{S}$, the bisimulation error $\Delta_\phi^\pi(s_i, s_j)$ can be upper-bounded by a sum of expected bisimulation Bellman residuals $\epsilon_\phi^\pi$:*

$$\Delta_\phi^\pi(s_i, s_j) \leq \frac{1}{1-\gamma} \mathbb{E}_{(s_i', s_j') \sim \mu_\pi} \left[ \epsilon_\phi^\pi(s_i', s_j') \right]. \tag{5}$$

Thereafter, the bisimulation Bellman residual is used as a surrogate objective to approximate the fixed point $G_\sim^\pi$ when learning our state representation. Indeed, the minimization of the bisimulation Bellman residual objective over all pairs $(s_i', s_j') \sim \mu_\pi$ leads to the minimization of the corresponding bisimulation error. This ensures that if the expected on-policy bisimulation Bellman residual (*i.e.,*

---

[3]For readability, we will conflate the notations $G^\pi$ and $G^\pi(x, y)$, they are the same if not specified

$\mathbb{E}_{\mu_\pi}[\epsilon_\phi^\pi]$, and we will use the term "expected bisimulation residual" in following) minimization objective is zero, then the bisimulation error must be zero for the state pairs under the same policy. However, when the dataset is limited, rather than an infinite transition set covering the whole MDP, minimizing the expected bisimulation residual will no longer be sufficient to guarantee a zero bisimulation error.

---

**Proposition 4.** *(The expected bisimulation residual is not sufficient over incomplete datasets).*
*If there exists states $s_i'$ and $s_j'$ not contained in dataset $\mathcal{D}$, where the occupancy $\mu_\pi(s_i'|s_i, a_i) > 0$ and $\mu_\pi(s_j'|s_j, a_j) > 0$ for some $(s_i, s_j) \sim \mu_\pi$, then there exists a bisimulation measurement $G_\phi^\pi$ and a constant $C > 0$ such that*

- *For all $(\hat{s}_i, \hat{s}_j) \in \mathcal{D}$, the bisimulation Bellman residual $\epsilon_\phi^\pi(\hat{s}_i, \hat{s}_j) = 0$.*
- *There exists $(s_i, s_j) \in \mathcal{D}$, such that the bisimulation error $\Delta_\phi^\pi(s_i, s_j) = C$.*

---

As an example, if we only have $(s_i, a_i, r, s_i')$ and $(s_j, a_j, r, s_j')$ in a dataset, where both rewards equal to zero for state $s_i$ and $s_j$, and if we choose $G_\phi^\pi(s_i, s_j) = C$, and $G_\phi^\pi(s_i', s_j') = \frac{1}{\gamma}C$, then the bisimulation Bellman residual is $\epsilon_\phi^\pi(s_i, s_j) = 0$, while the bisimulation error $\Delta_\phi^\pi = G_\phi^\pi(s_i, s_j) - 0 = C$ is strictly positive. Note that this failure case does not involve modifying the environment in an extremely adversarial manner, it simply occurs when we are required to estimate the representation of states with subsequent states that are missing from the dataset. Since the distance between the missing states can be arbitrarily large as they are out-of-distribution, directly minimizing the Bellman bisimulation error could achieve the minimal Bellman bisimulation error over the dataset, while not necessarily improving the state representation.

In the context of Offline RL, since the dataset is finite, bisimulation-based representation learning ought to be conceptualized as a pretraining process over the behavior policy $\pi_\beta$ of the dataset $\mathcal{D}$. However, the failure case above indicates that applying the bisimulation operator $\mathcal{F}^{\pi_\beta}$ and minimizing the associated Bellman bisimulation error does not necessarily ensure the sufficiency of the learned representation for downstream tasks. Ideally, if we had access to the fixed-point measurement $G_\sim^{\pi_\beta}$, then we could directly minimize the error between the approximation $G$ and the fixed-point $G_\sim^{\pi_\beta}$. However, given the static and incomplete nature of the dataset, acquiring the fixed-point $G_\sim^{\pi_\beta}$ explicitly is not feasible. From another perspective, the failure stems from out-of-distribution estimation errors. Assuming we could estimate the bisimulation exclusively with *in-sample learning*, this issue could be intuitively mitigated. As such, we resort to expectile regression as a regularizer, allowing us to circumvent the need for out-of-sample / unseen state pairs.

## 5 Method

In this section, we describe how we adapt existing bisimulation-based representation approaches to offline RL. We use the expectile-based operator to learn state representations that optimize the behavior measurement over the dataset, while avoiding overfitting to the incomplete data. In addition, we analyze the impact of reward scaling and propose as a consequence to normalize the reward difference in the bisimulation Bellman residual in order to satisfy the specific nature of different instantiations of the bisimulation measurement while keeping a lower value error. The pseudo-code of our method is shown in Algorithms in Appendix B.

### 5.1 Expectile-based Bisimulation Operator

The efficacy of expectile regression in achieving *in-sample learning* has already been demonstrated in previous research [28, 36]. Consequently, we will first describe our proposed *expectile*-based operator, and subsequently show how expectile regression can effectively address the aforementioned challenge. Specifically, we consider the update operator as follows:

$$
\left(\mathcal{F}_\tau^{\pi_\beta} G_\phi^{\pi_\beta}\right)(s_i, s_j) := \underset{G_\phi^{\pi_\beta}}{\arg\min} \, \mathbb{E}_{a_i \sim \pi_\beta(\cdot|s_i), a_j \sim \pi_\beta(\cdot|s_j)} \left[ \tau[\hat{\epsilon}]_+^2 + (1-\tau)[-\hat{\epsilon}]_+^2 \right],
$$

$$
\hat{\epsilon} = \mathbb{E}_{\substack{s_i' \sim T_{s_i}^{\pi_\beta} \\ s_j' \sim T_{s_j}^{\pi_\beta}}} \Big[ \underbrace{|r(s_i, a_i) - r(s_j, a_j)| + \gamma G_{\bar{\phi}}^{\pi_\beta}(s_i', s_j')}_{\text{target } G} - G_\phi^{\pi_\beta}(s_i, s_j) \Big],
\tag{6}
$$

where $\hat{\epsilon}$ is the estimated one-step bisimulation Bellman residual, $\pi_\beta$ is the behavior policy, $G_{\bar{\phi}}$ is the target encoder, updated using an exponential moving average, and $[\cdot]_+ = \max(\cdot, 0)$. Since the expectile operator in Equation 6 does not have a closed-form solution, in practice, we minimize it through gradient descent steps:

$$G_\phi^{\pi_\beta}(s_i, s_j) \leftarrow G_\phi^{\pi_\beta}(s_i, s_j) - 2\alpha \mathbb{E}_{a_i \sim \pi_\beta(\cdot|s_i), a_j \sim \pi_\beta(\cdot|s_j)} [\tau[\hat{\epsilon}]_+ + (1-\tau)[\hat{\epsilon}]_-] \tag{7}$$

where $\alpha$ is the step size. The fixed-point of the measurement obtained using this expectile-based operator is denoted as $G_\tau$. Although the utilization of the *expectile* statistics is well established, its application for estimating bisimulation measurement is not particularly intuitive. In the following, we will show how expectile-based operator can be helpful in addressing the aforementioned issue. First, it is worth noting that when $\tau = 1/2$, this operator becomes the bisimulation expectation of the behavior policy, *i.e.*, $\mathbb{E}_{\mu_{\pi_\beta}}[\hat{\epsilon}]$. Next, we shall consider how this operator performs when $\tau \to 1$. We show that under certain assumptions, our method indeed approximates an "optimal" measurement in terms of the given dataset. We first prove a technical lemma stating that the update operator is still a contraction, and then prove a lemma relating different expectiles, finally we derive our main result regarding the "optimality" of our method.

**Lemma 5.** *For any $\tau \in [0, 1)$, $\mathcal{F}_\tau^\pi$ is a $\gamma_\tau$-contraction, where $\gamma_\tau = 1 - 2\alpha(1-\gamma) \min\{\tau, 1-\tau\} < 1$.*

**Lemma 6.** *For any $\tau, \tau' \in [0, 1)$ with $\tau' \geq \tau$, and for all $s_i, s_j \in \mathcal{S}$ and any $\alpha$, we have $G_{\tau'} \geq G_\tau$.*

**Theorem 7.** *In deterministic MDP and fixed finite dataset, we have:*

$$\lim_{\tau \to 1} G_\tau(s_i, s_j) = \max_{\substack{a_i \in \mathcal{A}, a_j \in \mathcal{A} \\ s.t. \ \pi_\beta(a_i|s_i)>0, \pi_\beta(a_j|s_j)>0}} G_\sim^*((s_i, a_i), (s_j, a_j)). \tag{8}$$

*where $G_\sim^*((s_i, a_i), (s_j, a_j))$ is a fixed-point measurement constrained to the dataset and defined on the state-action space $\mathcal{S} \times \mathcal{A}$ as*

$$G_\sim^*((s_i, a_i), (s_j, a_j)) = |r(s_i, a_i) - r(s_j, a_j)| + \gamma \mathbb{E}_{\substack{s_i' \sim T_{s_i}^{\pi_\beta} \\ s_j' \sim T_{s_j}^{\pi_\beta}}} \left[ \max_{\substack{a_i' \in \mathcal{A}, a_j' \in \mathcal{A} \\ s.t. \ \pi_\beta(a_i'|s_i')>0, \pi_\beta(a_j'|s_j')>0}} G_\sim^*((s_i', a_i'), (s_j', a_j')) \right].$$

Intuitively, $G_\sim((s_i, a_i), (s_j, a_j))$ can be interpreted as a state-action value function $Q(\tilde{s}, \tilde{a})$ in a lifted MDP $\widetilde{M}$, and $G_\sim(s_i, s_j)$ as a state value function $V(\tilde{s})$. We defer the detailed explanation to Appendix E.

Theorem 7 illustrates that, as $\tau \to 1$, we are effectively approximating the maximum $G_\sim((s_i, a_i), (s_j, a_j))$ over actions $a_i', a_j'$ from the dataset. When we set $\tau = 1$, the expectile-based bisimulation operator achieves fully in-sample learning: we only consider state pairs that have corresponding actions in the dataset. For instance, only when we have $(s_i', a_i') \in \mathcal{D}$ and $(s_j', a_j') \in \mathcal{D}$, can we apply the measurement of $G_\sim^*$. As such, by manipulating $\tau$, we balance a trade-off between minimizing the expected bisimulation residual (for $\tau = 0.5$) and evaluating $G_\sim^*((s_i, a_i), (s_j, a_j))$ solely on the dataset (for $\tau = 1$), thereby sidestepping the failure case outlined in Proposition 4 in an implicit manner.

## 5.2 Reward Scaling

Most previous works [5, 54, 6, 52] have overlooked the impact of reward scaling in the bisimulation operator. To demonstrate its importance, we investigate a more general form of the bisimulation operator in Equation 2, given as:

$$\mathcal{F}^\pi G(s_i, s_j) = c_r \cdot |r_{s_i}^\pi - r_{s_j}^\pi| + c_k \cdot \mathbb{E}_{s_i', s_j'}^\pi [G(s_i', s_j')]. \tag{9}$$

We then can derive the following:

$$\begin{aligned}
G_\sim^\pi(s_i, s_j) = \mathcal{F}^\pi G_\sim^\pi(s_i, s_j) &= c_r \cdot |r_{s_i}^\pi - r_{s_j}^\pi| + c_k \cdot \mathbb{E}_{s_i', s_j'}^\pi [G_\sim^\pi(s_i', s_j')] \\
&\leq c_r \cdot (R_{\max} - R_{\min}) + c_k \cdot \mathbb{E}_{s_i', s_j'}^\pi [G_\sim^\pi(s_i', s_j')] \\
&\leq c_r \cdot (R_{\max} - R_{\min}) + c_k \cdot \max_{s_i', s_j'} G_\sim^\pi(s_i', s_j').
\end{aligned} \tag{10}$$

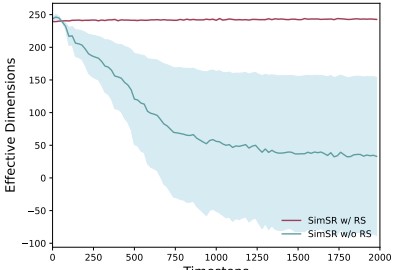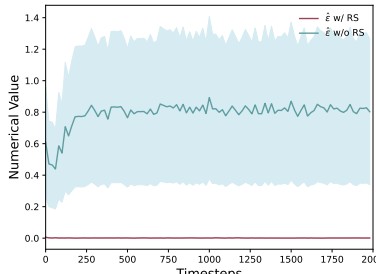

Figure 1: The effectiveness of Reward Scaling (**RS**) in SimSR on halfcheetah-medium-expert-v2, with results averaged on 3 random seeds. (**Left**) Effective Dimension [53] comparison: without **RS**, there is a significant reduction in the effective dimension, accompanied by a marked increase in instability as training progresses. (**Right**) Numerical value comparison of estimated bisimulation Bellman residual: $\hat{\epsilon}$ is persistently greater than 0 in the absence of **RS**, which indicates that target $G$ is invariably larger than $G_\phi$, suggesting that $G_\phi$ does not achieve steady convergence.

Accordingly, we have $G_\sim^\pi(s_i, s_j) \leq \frac{c_r \cdot (R_{\max} - R_{\min})}{1 - c_k}$. Adopting the conventional settings of $c_r = 1$ and $c_k = \gamma$ as suggested in [6, 52], could possibly result in a relatively large upper bound of $G_\sim^\pi$ between states. This is due to the common practice of setting $\gamma$ at 0.99. However, when bisimulation operators are instantiated with bounded distances, e.g., cosine distance, such a setting may be unsuitable. Therefore, it becomes important to tighten the upper bound.

Besides, we can also derive the value bound between the ground truth value function and the approximated value function:

**Theorem 8.** *(Value bound based on on-policy bisimulation measurements in terms of approximation error). Given an MDP $\widetilde{\mathcal{M}}$ constructed by aggregating states in an $\omega$-neighborhood, and an encoder $\phi$ that maps from states in the original MDP $\mathcal{M}$ to these clusters, the value functions for the two MDPs are bounded as*

$$\left| V^\pi(s) - \widetilde{V}^\pi(\phi(s)) \right| \leq \frac{2\omega + \hat{\Delta}}{c_r(1 - \gamma)}. \tag{11}$$

*where $\hat{\Delta} := \|\hat{G}_\sim^\pi - \hat{G}_\phi^\pi\|_\infty$ is the approximation error.*

In essence, Equation 10 and Theorem 8 reveal that: (i) there is a positive correlation between the reward scale $c_r$ and the upper bound of the fixed-point $G_\sim^\pi$, and (ii) a larger reward scale $c_r$ facilitates a more accurate approximation of the value function $\widetilde{V}^\pi(\phi(s))$ to its ground-truth value $V^\pi(s)$. It is important to note that $c_r$ also impacts the value of $\hat{\Delta}$, as depicted in Figure 7(Right)[4]. Therefore, it is crucial to first ensure the alignment with the instantiation of the bisimulation measurement, and then choose the largest possible $c_r$ to minimize the value error. For instance, as the SimSR operator [52] uses the cosine distance, $c_k = \gamma$ is predetermined. We should thus set $c_r \in [0, 1 - \gamma]$, and apply min-max normalization to the reward function. This can make $G_\sim^\pi \leq 1$ and therefore be consistent with the maximum value of 1 of the cosine distance. To achieve a tighter bound in Equation 11, we should then maximize the reward scale, setting $c_r$ to $1 - \gamma$. Figure 7 illustrates the effectiveness of this reward scaling.

## 6 Experiments

### 6.1 Performance Comparison in D4RL Benchmark

**Implementation Details** We analyze our proposed method on the D4RL benchmark [15] of OpenAI gym MuJoCo tasks [50] which includes a variety of datasets that have been commonly used in the

---

[4]Despite Figure 7(Right) depicting the approximate residual $\hat{\epsilon}$, we have drawn a connection between $\epsilon_\phi^\pi$ and $\Delta_\phi^\pi$ in the Appendix, which can reflect the possible situations for $\hat{\Delta}$.

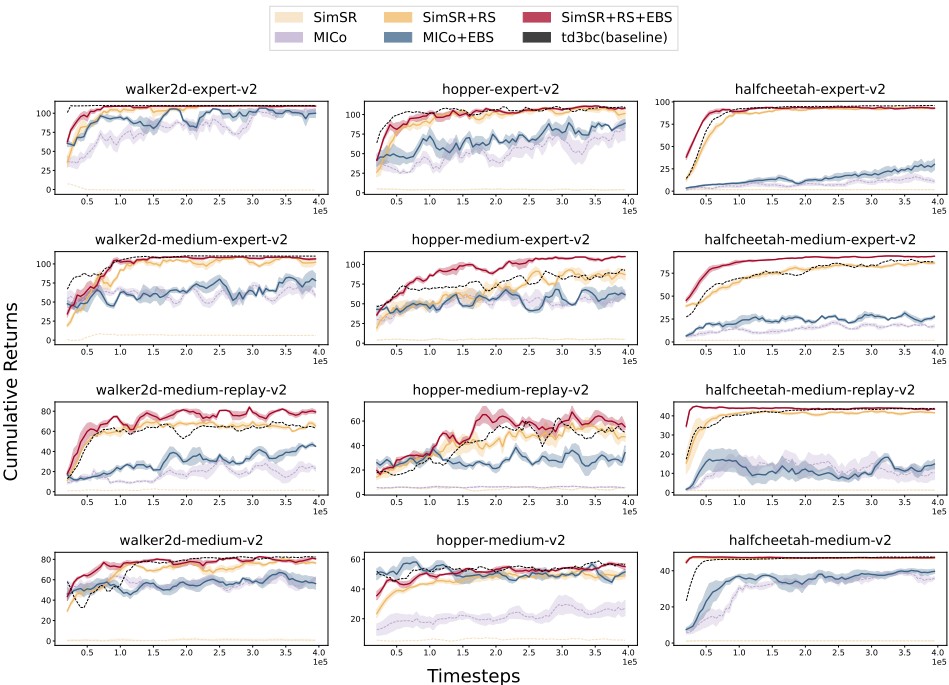

Figure 2: Performance comparison on 12 D4RL tasks over 10 seeds with one standard error shaded in the default setting. For every seed, the average return is computed every 10,000 training steps, averaging over 10 episodes. The horizontal axis indicates the number of transitions trained on. The vertical axis indicates the normalized average return.

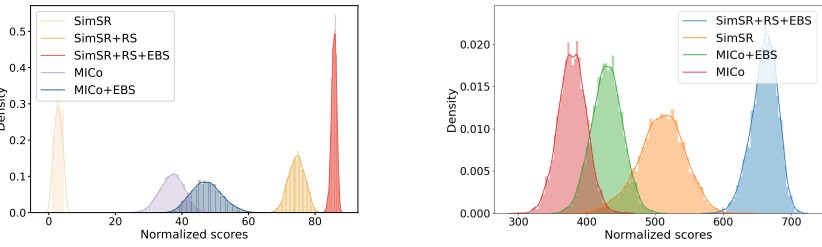

Figure 3: Bootstrapping distributions for uncertainty in IQM (*i.e.* inter-quartile mean) measurement on D4RL tasks (left) and visual D4RL tasks (right), following from the performance criterion in [2].

Offline RL community. To illustrate the effectiveness of our method, we implement it on top of two bisimulation-based approaches, **MICo** [6] and **SimSR** [52]. It is worth noting that there are two versions of SimSR depending on its use of a latent dynamics model: SimSR_basic follows the dynamics that the environment provides, and SimSR_full constructs latent dynamics for sampling successive latent states. We opt for SimSR_basic as our backbone, as it exhibits superior and more stable performance in the D4RL benchmark tasks compared to SimSR_full. Additionally, to explore the impact of bisimulation-based representation learning on the downstream performance of policy learning, we build these approaches on top of the Offline RL method **TD3BC** [16]. We examine three environments: halfcheetah, hopper, and walker2d, with four datasets per task: expert, medium-expert, medium-replay, and medium. We first pretrain the encoder during $100k$ timesteps, then freeze it, pass the raw state through the frozen encoder to obtain the representations that serve as input for the Offline RL algorithm. Further details on the experiment setup are included in Appendix F.

**Analysis** Figure 2 illustrates the performance of two approaches and their variants in the D4RL tasks. We use ***EBS*** to represent the scheme of employing the expectile-based operator, while ***RS***

denotes the reward scaling scheme. The latter includes both min-max reward normalization and penalization coefficient with $(1-\gamma)$ in the bisimulation operator. As discussed in Section 5.2, the role of reward scaling varies depending on the specific instantiation of $G^5$. We observe that without **RS**, SimSR almost fails in every dataset, which aligns with our understanding of the critical role reward scaling plays. The results also illustrate that **EBS** effectively enhances the downstream performance of the policy for both SimSR and MICo. It is noteworthy that in this experiment, we set $\tau = 0.6$ for the expectile in SimSR and $\tau = 0.7$ in MICo across all datasets, demonstrating the robustness of this hyperparameter. Regarding SimSR, when **RS** is applied (**SimSR+RS**), the performance is comparable to the TD3BC baseline, while the incorporation of the expectile-based operator (**SimSR+RS+EBS**) further enhances final performance and sample efficiency. Besides, we additionally present the IQM normalized return of all variants in Figure 3, illustrating our performance gains over the backbones. Further, we have also constructed an ablation study to investigate the impact of different settings of $\tau$, the results show that a suitable expectile $\tau$ is crucial for control tasks. We present the corresponding results in Appendix E.

## 6.2 Performance Comparison in V-D4RL Benchmark

**Implementation details** We also evaluate our method on a visual observation setting of DM-Control suite (DMC) tasks, V-D4RL benchmark [35]. Similar to the previous experiment, we add the proposed schemes on top of MICo and SimSR. In the experiments, we notice that the latent dynamics modeling can help to boost performance for the visual setting, hence we use SimSR_full as the backbone. Additionally, we also notice that MICo often gives really poor performance in the V-D4RL benchmark, while adding latent dynamics alleviates the issue. Therefore, we boost MICo with explicit dynamics modeling for a fair comparison. To compare the performance with the other representation approaches, we include 4 competitive representation learning approaches for Offline RL, including DRIML [38], HOMER [39],

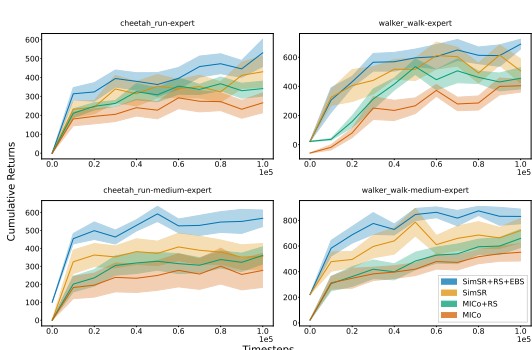

Figure 4: Performance comparison on V-D4RL benchmark.

CURL [31], and Inverse model [44]. Detailed descriptions of these approaches can be found in Appendix G.

**Analysis** We evaluate all aforementioned approaches by integrating the pre-trained encoder from each into an Offline RL method DrQ+BC [35], which combines data augmentation techniques with TD3BC. The results in Table 1 and Figure 4 illustrate the effectiveness of our proposed method, the numerical improvements are underlined with red upward arrows. Compared to the other baselines, while **SimSR+RS+EBS** does not achieve the highest score in all datasets, it achieves the best overall performance. Besides, our modifications on MICo and SimSR consistently show significant improvements. This indicates that our proposed method is not only applicable to raw-state inputs but also compatible with pixel-based observations.

## 7 Discussion

**Limitations and Future Work** While $\tau$ remains constant in our D4RL experiments, optimal performance may arise under different $\tau$ settings, contingent on the specific attributes of the dataset. Therefore, to yield the best outcomes, one might need to set various $\tau$ to identify the most suitable value. However, this process could consume substantial computational resources. Another area of potential study involves evaluating the effectiveness of our approach in off-policy settings, given that off-policy settings may also lead to similar failure cases.

---

[5] Since MICo does not necessitate a particular upper bound, RS may be harmful to its performance. Our experiments have substantiated this observation, leading us to exclude the MICo+RS results from Figure 2.

Table 1: Performance comparison with several other baselines on V-D4RL benchmark, averaged on 3 random seeds.

| Dataset | CURL | DRIMLC | HOMER | ICM | MICo → MICo+EBS | SimSR → SimSR+RS+EBS |
|---|---|---|---|---|---|---|
| cheetah-run-medium | 392 | **524** | 475 | 365 | 177 → 449 (↗ 272) | 391 → 491(↗ 100) |
| walker-walk-medium | 452 | 425 | 439 | 358 | 450 → 447 (—) | 443 → **480**(↗ 37) |
| cheetah-run-medium-replay | 271 | 395 | 306 | 251 | 335 → 357 (↗ 22) | 374 → **462**(↗ 88) |
| walker-walk-medium-replay | 265 | 235 | **283** | 167 | 207 → 240 (↗ 33) | 197 → 240(↗ 43) |
| cheetah-run-medium-expert | 348 | 403 | 383 | 280 | 282 → 341 (↗ 59) | 360 → **547**(↗ 187) |
| walker-walk-medium-expert | 729 | 399 | 781 | 606 | 586 → 635(↗ 49) | 755 → **845**(↗ 90) |
| cheetah-run-expert | 200 | 310 | 218 | 237 | 308 → 331(↗ 23) | 409 → **454**(↗ 45) |
| walker-walk-expert | 769 | 427 | 686 | **850** | 370 → 447 (↗ 77) | 578 → 580 (—) |
| total | 3426 | 3118 | 3571 | 3114 | 2715 → 3253 (↗ 538) | 3507 → **4043** (↗ 536) |

**Conclusion**    In this work, we highlight the effectiveness of the bisimulation operator over incomplete datasets and emphasize the crucial role of reward scaling in Offline settings. By employing the expectile operator in bisimulation, we manage to strike a balance between behavior measurement and greedy assignment of the measurement over datasets. We also propose a reward scaling strategy to reduce the risk of representation collapse in specific bisimulation-based measurements. Empirical studies show the effectiveness of our proposed modifications.

# Acknowledgments

This work was partially supported by the NSFC under Grants 92270125 and 62276024, as well as the National Key R&D Program of China under Grant No.2022YFC3302101.

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

# Appendix

## Contents

# A   Notation

Table 2 summarizes our notation.

Table 2: Table of Notation.

| Notation | Meaning | Notation | Meaning |
|----------|---------|----------|---------|
| $\mathcal{M}$ | MDP | $\widetilde{\mathcal{M}}$ | Lifted MDP (auxiliary MDP) |
| $\mathcal{S}$ | state space | $\mathcal{A}$ | action space |
| $T$ | transition function | $r$ | reward function |
| $\gamma$ | discount factor | $\pi$ | policy of the agent |
| $V^\pi(s)$ | state value function given policy $\pi$ | $\mathcal{D}$ | dataset |
| $\pi_\beta$ | behavior policy | $\mu_\beta(s)$ | state occupancy of the dataset |
| $\mathcal{F}^\pi$ | on-policy bisimulation operator | $g_\sim^\pi$ | $\pi$-bisimulation metric |
| $D(\cdot,\cdot)$ | a specific distance | $G_\sim^\pi$ | fixed point of MICo and SimSR |
| $\phi$ | state encoder | $G_\phi^\pi(s_i,s_j)$ | parameterized bisimulation measurement |
| $\Delta_\phi^\pi$ | bisimulation error | $\epsilon_\phi^\pi$ | bisimulation Bellman residual |
| $\mu_\pi(s)$ | stationary distribution over states on policy $\pi$ | $\mu_\pi$ | the distribution over pairs of states |
| $\mathbb{E}_{\mu_\pi}[\epsilon_\phi^\pi]$ | expected on-policy bisimulation Bellman residual | $\mathcal{F}^{\pi_\beta}$ | behavior bisimulation operator |
| $\tau$ | expectile term | $\gamma_\tau$ | discount factor with expectile |
| $\mathcal{F}_\tau^{\pi_\beta}$ | behavior bisimulation operator with expectile | $\hat{\epsilon}$ | estimated one-step residual |
| $G_{\bar{\phi}}$ | bisimulation measurement parameterized by target encoder | $G_\sim(s_i,a_i,s_j,a_j)$ | a measurement on state-action space |
| $G_\sim^*(s_i,a_i,s_j,a_j)$ | maximum measurement constrained to dataset | $c_r$ | scale term of reward in bisimulation |
| $c_k$ | scale term of transition in bisimulation | $\widetilde{V}^\pi(\phi(s))$ | value function based on state encoder |
| $\omega$ | distance bound of aggregating neighbor | $\hat{\Delta}$ | approximation error of bisimulation measurement |

# B   Algorithm

We provide the algorithm in Algorithm 1, and a pytorch-like implementation build on top of SimSR in Algorithm 2.

---

**Algorithm 1** Proposed Implementation

---

1: **Stage 1 Preprocessing:**
2: Min-Max reward normalization: $\bar{r} = \frac{r - r_{\min}}{r_{\max} - r_{\min}}$
3: **Stage 2 Pretraining the encoder:**
4: Initialize encoder parameter $\phi$, expectile $\tau$, learning rate $\alpha$, discount factor $\gamma$.
5: **for** each gradient step **do**
6:     Apply reward scaling when computing $\hat{\epsilon}$:

$$\hat{\epsilon} = (1-\gamma)|\bar{r}(s_i,a_i) - \bar{r}(s_j,a_j)| + \gamma G_{\bar{\phi}}^{\pi_\beta}(s_i',s_j') - G_\phi^{\pi_\beta}(s_i,s_j) \tag{12}$$

7:     Update encoder $\phi$:

$$\phi \leftarrow \phi - 2\alpha \mathbb{E}_{a_i \sim \pi_\beta(\cdot|s_i), a_j \sim \pi_\beta(\cdot|s_j)} [\tau[\hat{\epsilon}]_+ + (1-\tau)[\hat{\epsilon}]_-] \tag{13}$$

8: **end for**
9: **Stage 3 Training value function and policy network:**
10: Initialize value function parameter $\psi$, policy network parameter $\theta$, learning rate $\lambda_V$ and $\lambda_\pi$.
11: **for** each gradient step **do**
12:     Sample tuple $(s,a,s',\bar{r})$ from dataset $\mathcal{D}$
13:     Encode the states to representation space: $z = \phi(s), z' = \phi(s')$
14:     Update value function with $(z,a,z',\bar{r})$:

$$\psi = \psi - \lambda_V \nabla_\psi \mathcal{L}_V(\psi). \tag{14}$$

15:     Update policy network with $(z,a,z',\bar{r})$:

$$\theta = \theta - \lambda_\pi \nabla_\theta \mathcal{L}_\pi(\theta). \tag{15}$$

16: **end for**

---

**Algorithm 2** SimSR+RS+EBS Pseudocode, PyTorch-like

```python
class ReplayBuffer(object):
    def __init__(self):
        ...
        self.reward_normalization()
        ...
    ...
    def reward_normalization(self):
        r_max = self.reward.max()
        r_min = self.reward.min()
        self.reward = (self.reward - r_min) / (r_max - r_min)

def compute_distance(features_a, features_b):
    similarity_matrix = torch.matmul(features_a, features_b.T)
    dis = 1-similarity_matrix
    return dis

def expectile_loss(diff, expectile):
    weight = torch.where(diff > 0, expectile, (1 - expectile))
    return weight * (diff ** 2)

# encoder: mlp, encoder network, the output is l2-normalized
# target_encoder: mlp, same as encoder, updated by EMA
# discount: discount factor $\gamma$
# slope: expectile $\tau$
def compute_ebs_loss(encoder, target_encoder, replay_buffer, batch_size, discount, slope):
    observation, action, reward, discount, next_observation = replay_buffer.sample(batch_size) # sample a
        batch of tuples from replay buffer
    latent_state = encoder(observation)
    latent_next_state = target_encoder(next_observation)
    r_diff = (1 - discount) * torch.abs(reward.T - reward)
    next_diff = compute_distance(latent_next_state, latent_next_state)
    z_diff = compute_distance(latent_state, latent_state)
    bisimilarity = r_diff + discount * next_diff

    encoder_loss = expectile_loss(bisimilarity.detach() - z_diff, slope)
    encoder_loss = encoder_loss.mean()
    return encoder_loss
```

# C   Technical backgrounds

## C.1   Bisimulation metric

Bisimulation measures equivalence relations on MDPs with a recursive form: two states are deemed equivalent if they share the equivalent distributions over the next equivalent states and they have the same immediate reward [30, 20]. However, since bisimulation considers equivalence for all actions, including bad ones, it commonly results in "pessimistic" outcomes. Instead, [5] developed $\pi$-bisimulation which removes the requirement of considering each action and only needs to consider the actions induced by a policy $\pi$.

**Definition 9.** *[5] Given an MDP $\mathcal{M}$, an equivalence relation $E^\pi \subseteq \mathcal{S} \times \mathcal{S}$ is a $\pi$-bisimulation relation if whenever $(\mathbf{s}, \mathbf{u}) \in E^\pi$ the following properties hold:*

1. $r(s, \pi) = r(u, \pi)$

2. $\forall C \in \mathcal{S}_{E^\pi}, T(C|s, \pi) = T(C|u, \pi)$

*where $\mathcal{S}_{E^\pi}$ is the state space $\mathcal{S}$ partitioned into equivalence classes defined by $E^\pi$. Two states $s, u \in S$ are $\pi$-bisimilar if there exists a $\pi$-bisimulation relation $E^\pi$ such that $(s, u) \in E^\pi$.*

However, $\pi$-bisimulation is still too stringent to be applied at scale as $\pi$-bisimulation relation emphasizes the equivalence is a binary property: either two states are equivalent or not, thus becoming too sensitive to perturbations in the numerical values of the model parameters. The problem becomes even more prominent when deep frameworks are applied.

Thereafter, they proposed a $\pi$-bisimulation metric to leverage the absolute value between the immediate rewards w.r.t. two states and the 1-Wasserstein distance ($\mathcal{W}_1$) between the transition distributions conditioned on the two states and the policy $\pi$ to formulate such measurement:

**Theorem 10.** *Define $\mathcal{F}^\pi : \mathbb{M} \to \mathbb{M}$ by $\mathcal{F}^\pi(d)(s, u) = |R_s^\pi - R_u^\pi| + \gamma \mathcal{W}_1(d)(T_s^\pi, T_u^\pi)$, then $\mathcal{F}^\pi$ has a least fixed point $d_\sim^\pi$, and $d_\sim^\pi$ is a $\pi$-bisimulation metric.*

Although the Wasserstein distance is a powerful metric to calculate the distance between two probability distributions, it requires to enumerate all states which is impossible in RL tasks of continuous state space. Various extensions have been proposed [54, 6, 52] to reduce the computational complexity. DBC [54] extend bisimulation metrics to learn state representation, via minimizing the $\ell_1$-norm distance of representations and the bisimulation metrics, meanwhile modeling the latent dynamics as Gaussian and utilizing $W_2$ distance to compute it, which can be formulated as a closed-form result. However, DBC has several issues like loss function mismatch and specific requirements for Gaussian modeling, which limits its application and performance.

## C.2  MICo distance

MICo distance [6], tackles the above issue by restricting the coupling class to the independent coupling to avoid intractable Wasserstein distance computation. The MICo operator and its associated theoretical guarantee are given as:

**Theorem 11.** *[6] Given a policy $\pi$, MICo distance $\mathcal{F}^\pi$ is defined as:*

$$\mathcal{F}^\pi U(s_i, s_j) = |r^\pi_{s_i} - r^\pi_{s_j}| + \gamma \mathbb{E}_{s'_i \sim T^\pi_s, s'_j \sim T^\pi_{s_j}}[U(s'_i, s'_j)] \tag{16}$$

*has a fixed point $U^\pi$.*

By considering the Wasserstein distance in the definition of bisimulation metrics can be upper-bounded by taking a restricted class of couplings of the transition distributions, MICo restricts the coupling class precisely to the singleton containing the independent coupling, utilizing the Independent Couple sampling strategy to bypass the computation of the Wasserstein distance. However, MICo distance $U$ requires to be a Łukaszyk-Karmowski metric, which does not satisfy the identity of indiscernibles. As a result, the approximated distance on the learned embedding space based on the MICo distance, which involves a Łukaszyk-Karmowski metric to measure the distance between dynamics, may suffer from the violation issue of the identity of indiscernibles.

## C.3  SimSR operator

To avoid the potential representation collapse, SimSR [52] develop a more concise update operator to learn state representation more effectively. Coupling with cosine distance, SimSR defines its operator as:

**Theorem 12.** *[52] Given a policy $\pi$, Simple State Representation (SimSR) is updated as:*

$$\mathcal{F}^\pi \overline{cos}_\phi(s_i, s_j) = |r^\pi_{s_i} - r^\pi_{s_j}| + \gamma \mathbb{E}_{s'_i \sim T^\pi_{s_i}, s'_j \sim T^\pi_{s_j}}[\overline{cos}_\phi(s'_i, s'_j)] \tag{17}$$

*has the same fixed point as MICo.*

Further, considering the latent dynamics can be beneficial to representation learning, they additionally develop a form of operator including dynamics modeling:

**Theorem 13.** *[52] Given a policy $\pi$, and a latent dynamics model $\hat{T}$, SimSR is updated as*

$$\mathcal{F}^\pi \overline{cos}_\phi(s_i, s_j) = |r^\pi_{s_i} - r^\pi_{s_j}| + \gamma \mathbb{E}_{z'_i \sim \hat{T}^\pi_{\phi(s_i)}, z'_j \sim \hat{T}^\pi_{\phi(s_j)}}[\overline{cos}(z'_i, z'_j)]. \tag{18}$$

*If latent dynamics are specified, $\mathcal{F}^\pi$ has a fixed point.*

When considering MICo distance and the basic version of SimSR, we can notice that they have a similar recursive iteration formulation. And therefore both works can be generalized under:

$$\mathcal{F}^\pi G^\pi(s_i, s_j) = |r^\pi_{s_i} - r^\pi_{s_j}| + \gamma \mathbb{E}_{\substack{s'_i \sim T^\pi_{s_i} \\ s'_j \sim T^\pi_{s_j}}}[G^\pi(s'_i, s'_j)], \tag{19}$$

while the instantiation of $G$ varies in these two approaches.

## C.4  Lifted MDP

The connection between bisimulation-based operators and lifted MDP can be referred to [6]. We provide the corresponding Lemma here for reference.

**Lemma 2.** *(**Lifted MDP**) The bisimulation-based update operator $\mathcal{F}^\pi$ for $\mathcal{M}$, is the Bellman evaluation operator for a specific lifted MDP.*

*Proof.* Given the MDP specified by the tuple $(\mathcal{S}, \mathcal{A}, T, R)$, we construct a lifted MDP $(\widetilde{\mathcal{S}}, \widetilde{\mathcal{A}}, \widetilde{T}, \widetilde{R})$, by taking the state space to be $\widetilde{\mathcal{S}} = \mathcal{S}^2$, the action space to be $\widetilde{\mathcal{A}} = \mathcal{A}^2$, the transition dynamics to be given by $\widetilde{T}_{\tilde{s}}^{\tilde{a}}(\tilde{s}') = \widetilde{T}_{(s_i,s_j)}^{(a_i,a_j)}((s_i',s_j')) = T_{s_i}^{a_i}(s_i')T_{s_j}^{a_j}(s_j')$ for all $(s_i, s_j), (s_i', s_j') \in \mathcal{S}^2, a_i, a_j \in \mathcal{A}$, and the action-independent rewards to be $\widetilde{R}_{\tilde{s}} = \widetilde{R}_{(s_i,s_j)} = |r_{s_i}^\pi - r_{s_j}^\pi|$ for all $s_i, s_j \in \mathcal{S}$. The Bellman evaluation operator $\widetilde{\mathcal{F}}^{\tilde{\pi}}$ for this lifted MDP at discount rate $\gamma$ under the policy $\tilde{\pi}(\tilde{a}|\tilde{s}) = \tilde{\pi}(a_i, a_j|s_i, s_j) = \pi(a_i|s_i)\pi(a_j|s_j)$ is given by (for all $G^\pi \in \mathbb{R}^{\mathcal{S}\times\mathcal{S}}$ and $(s_i, s_j) \in \mathcal{S}\times\mathcal{S}$):

$$(\widetilde{\mathcal{F}}^{\tilde{\pi}}\tilde{G}^\pi)(\tilde{s}) = \widetilde{R}_{\tilde{s}} + \gamma \sum_{\tilde{s}'\in\widetilde{\mathcal{S}}} \widetilde{T}_{\tilde{s}}^{\tilde{a}}(\tilde{s}')\tilde{\pi}(\tilde{a}|\tilde{s})\tilde{G}^\pi(\tilde{s}')$$

$$(\widetilde{\mathcal{F}}^{\tilde{\pi}}G^\pi)(s_i, s_j) = \widetilde{R}_{(s_i,s_j)} + \gamma \sum_{(s_i',s_j')\in\mathcal{S}^2} \widetilde{T}_{(s_i,s_j)}^{(a_i,a_j)}((s_i',s_j'))\tilde{\pi}(a_i, a_j|s_i, s_j)G^\pi(s_i', s_j')$$

$$= |r_{s_i}^\pi - r_{s_j}^\pi| + \gamma \sum_{(s_i',s_j')\in\mathcal{S}^2} T_{s_i}^\pi(s_i')T_{s_j}^\pi(s_j')G^\pi(s_i', s_j') = (\mathcal{F}_M^\pi G^\pi)(s_i, s_j) . \square$$

### C.5 Expectile Regression

Expectile regression, a method in statistics, is an extension of quantile regression that provides a more detailed analysis of a distribution's tail. This technique aims to estimate the expectiles of a conditional distribution, which are like percentiles but with respect to the mean, not the median. In essence, expectile regression can help capture the structure of data variability and analyze extreme observations in a more precise manner than quantile regression. The $\tau \in (0, 1)$ expectile of some random variable $X$ is defined as a solution to the asymmetric least squares problem:

$$\underset{m_\tau}{\arg\min} \mathbb{E}_{x\sim X}\left[L_2^\tau(x - m_\tau)\right], \quad (20)$$

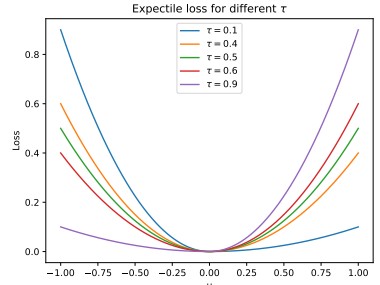

Figure 5: The asymmetric squared loss used for expectile regression. Larger $\tau$ gives more weight to positive differences.

where $L_2^\tau(u) = |\tau - \mathbb{1}(u < 0)|u^2$. That is, for $\tau > 0.5$, this asymmetric loss function downweights the contributions of $x$ values smaller than $m_\tau$ while giving more weights to larger values. Figure 5 shows the illustration of this asymmetric loss. More detailed descriptions can be found in [28, 36].

## D   Proof

### D.1   Connection between bisimulation error and bisimulation Bellman residual

In this section, we will revise some definitions a bit for obtaining the equivalence between bisimulation error and bisimulation Bellman residual. We first define *bisimulation error* $\Delta_\phi^\pi$ that measure the distance of the approximation $G_\phi^\pi$ to the fixed point $G_\sim^\pi$ as:

$$\Delta_\phi^\pi := G_\phi^\pi(s_i, s_j) - G_\sim^\pi(s_i, s_j). \quad (21)$$

And define *bisimulation Bellman residual* $\epsilon_\phi^\pi$ as:

$$\epsilon_\phi^\pi := G_\phi^\pi(s_i, s_j) - \mathcal{F}^\pi G_\phi^\pi(s_i, s_j). \quad (22)$$

Notably, this is slightly different from the notation in Section 4 given the fact that we do not apply absolute value here. Then, we can have the following theorems.

**Theorem 14.** *(The bisimulation Bellman residual can be defined as a function of the bisimulation error)*

$$\epsilon_\phi^\pi(s_i, s_j) = \Delta_\phi^\pi(s_i, s_j) - \gamma \mathbb{E}_{\substack{s_i' \sim T_{s_i}^\pi \\ s_j' \sim T_{s_j}^\pi}} [\Delta_\phi^\pi(s_i', s_j')], \tag{23}$$

*Proof.* This follows directly from the bisimulation update operator:

$$
\begin{aligned}
\epsilon_\phi^\pi(s_i, s_j) &= G_\phi^\pi(s_i, s_j) - \mathcal{F}^\pi G_\phi^\pi(s_i, s_j) \\
&= G_\sim^\pi(s_i, s_j) + \Delta_\phi^\pi(s_i, s_j) - \mathcal{F}^\pi(G_\sim^\pi(s_i, s_j) + \Delta_\phi^\pi(s_i, s_j)) \\
&= \Delta_\phi^\pi(s_i, s_j) - \gamma \mathbb{E}_{\substack{s_i' \sim T_{s_i}^\pi \\ s_j' \sim T_{s_j}^\pi}} [\Delta_\phi^\pi(s_i', s_j')]
\end{aligned} \tag{24}
$$

$\square$

**Theorem 15.** *(The bisimulation error can be defined as a function of the bisimulation Bellman residual).* *For any state pair* $(s_i, s_j) \in \mathcal{S} \times \mathcal{S}$, *the approximation error* $\Delta_\phi^\pi(s_i, s_j)$ *can be defined as a function of the Bellman bisimulation error* $\epsilon_\phi$

$$\Delta_\phi^\pi(s_i, s_j) = \frac{1}{1-\gamma} \mathbb{E}_{(s_i', s_j') \sim \mu_\pi} \left[ \epsilon_\phi^\pi(s_i', s_j') \right]. \tag{25}$$

*Proof.* Our proof follows similar steps to the proof of Lemma 6.1 in [26] and Theorem 1 in [17]. First by definition:

$$
\begin{aligned}
\Delta_\phi^\pi(s_i, s_j) &:= G_\phi^\pi(s_i, s_j) - G_\sim^\pi(s_i, s_j) \\
&\Rightarrow G_\sim^\pi(s_i, s_j) = G_\phi^\pi(s_i, s_j) - \Delta_\phi^\pi(s_i, s_j)
\end{aligned} \tag{26}
$$

Then we can decompose the error:

$$
\begin{aligned}
\Delta_\phi^\pi(s_i, s_j) &= G_\phi^\pi(s_i, s_j) - G_\sim^\pi(s_i, s_j) \\
&= G_\phi^\pi(s_i, s_j) - \left( |r_{s_i}^\pi - r_{s_j}^\pi| + \gamma \mathbb{E}_{\substack{s_i' \sim T_{s_i}^\pi \\ s_j' \sim T_{s_j}^\pi}} [G_\sim^\pi(s_i', s_i')] \right) \\
&= G_\phi^\pi(s_i, s_j) - \left( |r_{s_i}^\pi - r_{s_j}^\pi| + \gamma \mathbb{E}_{\substack{s_i' \sim T_{s_i}^\pi \\ s_j' \sim T_{s_j}^\pi}} [G_\phi^\pi(s_i', s_j') - \Delta_\phi^\pi(s_i', s_j')] \right) \\
&= G_\phi^\pi(s_i, s_j) - \left( |r_{s_i}^\pi - r_{s_j}^\pi| + \gamma \mathbb{E}_{\substack{s_i' \sim T_{s_i}^\pi \\ s_j' \sim T_{s_j}^\pi}} [G_\phi^\pi(s_i', s_j')] \right) + \gamma \mathbb{E}_{\substack{s_i' \sim T_{s_i}^\pi \\ s_j' \sim T_{s_j}^\pi}} [\Delta_\phi^\pi(s_i', s_j')] \\
&= \epsilon_\phi^\pi(s_i, s_j) + \gamma \mathbb{E}_{\substack{s_i' \sim T_{s_i}^\pi \\ s_j' \sim T_{s_j}^\pi}} [\Delta_\phi^\pi(s_i', s_j')]
\end{aligned} \tag{27}
$$

By considering the operator $G$ as the Bellman evaluation operator for the lifted MDP (See Section C.4), we can rewrite the formula as:

$$\Delta_\phi^{\tilde\pi}(\tilde{x}) = \epsilon_\phi^{\tilde\pi}(\tilde{x}) + \gamma \mathbb{E}_{\tilde{x}' \sim T_{\tilde{x}}^{\tilde\pi}} \left[ \Delta_\phi^{\tilde\pi}(\tilde{x}') \right]. \tag{28}$$

Then we can treat $\Delta_\phi^{\tilde\pi}(\tilde{x})$ as a value function and $\epsilon_\phi^{\tilde\pi}(\tilde{x})$ as reward, we can see that:

$$\Delta_\phi^{\tilde\pi}(\tilde{x}) = \frac{1}{1-\gamma} \mathbb{E}_{\tilde{x}' \sim T_{\tilde{x}}^{\tilde\pi}} \left[ \epsilon_\phi^{\tilde\pi}(\tilde{x}') \right]. \tag{29}$$

Then we can obtain

$$\Delta_\phi^\pi(s_i, s_j) = \frac{1}{1-\gamma} \mathbb{E}_{(s_i', s_j') \sim \mu_\pi} \left[ \epsilon_\phi^\pi(s_i', s_j') \right]. \tag{30}$$

$\square$

## D.2 Thoerem 3

**Theorem 3.** *(Bisimulation error upper-bound).* *Let $\mu_\pi(s)$ denote the stationary distribution over states, let $\mu_\pi(\cdot, \cdot)$ denote the joint distribution over synchronized pairs of states $(s_i, s_j)$ sampled independently from $\mu_\pi(\cdot)$. For any state pair $(s_i, s_j) \in \mathcal{S} \times \mathcal{S}$, the bisimulation error $\Delta_\phi^\pi(s_i, s_j)$ can be upper-bounded by a sum of expected bisimulation Bellman residuals $\epsilon_\phi^\pi$:*

$$\Delta_\phi^\pi(s_i, s_j) \leq \frac{1}{1-\gamma} \mathbb{E}_{(s_i, s_j) \sim \mu_\pi} \left[ \epsilon_\phi^\pi(s_i, s_j) \right]. \tag{31}$$

*Proof.* We start from Equation 25 in Section D.1.

$$\begin{aligned}
\Delta_\phi^\pi(s_i, s_j) &= \frac{1}{1-\gamma} \mathbb{E}_{(s_i', s_j') \sim \mu_\pi} \left[ \epsilon_\phi^\pi(s_i', s_j') \right] \\
\Rightarrow |\Delta_\phi^\pi(s_i, s_j)| &= \frac{1}{1-\gamma} \left| \mathbb{E}_{(s_i', s_j') \sim \mu_\pi} \left[ \epsilon_\phi^\pi(s_i', s_j') \right] \right| \\
&\leq \frac{1}{1-\gamma} \mathbb{E}_{(s_i', s_j') \sim \mu_\pi} \left[ \left| \epsilon_\phi^\pi(s_i', s_j') \right| \right].
\end{aligned} \tag{32}$$

Then when we define bisimulation error $\Delta_\phi^\pi(s_i, s_j) := |\Delta_\phi^\pi(s_i, s_j)|$ and bisimulation Bellman residual $\epsilon_\phi^\pi(s_i', s_j') := |\epsilon_\phi^\pi(s_i', s_j')|$, we have

$$\Delta_\phi^\pi(s_i, s_j) \leq \frac{1}{1-\gamma} \mathbb{E}_{(s_i', s_j') \sim \mu_\pi} \left[ \epsilon_\phi^\pi(s_i', s_j') \right]. \tag{33}$$

$\square$

## D.3 Proposition 4

**Proposition 4.** *(The expected bisimulation residual is not sufficient over incomplete datasets).* *If there exists states $s_i'$ and $s_j'$ not contained in dataset $\mathcal{D}$, where the occupancy $\mu_\pi(s_i'|s_i, a_i) > 0$ and $\mu_\pi(s_j'|s_j, a_j) > 0$ for some $s_i \in \mathcal{D}, s_j \in \mathcal{D}$, then there exists a bisimulation measurement and $C > 0$ such that*

- *For all $(\hat{s}_i, \hat{s}_j) \in \mathcal{D}$, the bisimulation Bellman residual $\epsilon_\phi^\pi(\hat{s}_i, \hat{s}_j) = 0$.*

- *There exists $(s_i, s_j) \in \mathcal{D}$, such that the bisimulation error $\Delta_\phi^\pi(s_i, s_j) = C$.*

*Proof.* This is a direct consequence of Theorem 15. Let $\mathcal{D}'$ contain the set of state pairs $(s_i', s_j')$ not contained in the dataset $\mathcal{D}$, where the next-state pair occupancy $\mu_\pi(s_i', s_j'|s_i, a_i, s_j, a_j) > 0$. Let $\mathcal{D}_{\text{unique}}$ be the set of unique state pairs in $\mathcal{D}$. It follows that

$$\begin{aligned}
\Delta_\phi^\pi(s_i, s_j) =& \frac{1}{1-\gamma} \mathbb{E}_{(s_i', s_j') \sim \mu_\pi} \left[ \epsilon_\phi^\pi(s_i', s_j') \right] \\
=& \frac{1}{1-\gamma} \sum_{(s_i', s_j') \sim \mathcal{D}_{\text{unique}}} \mu_\pi((s_i', s_j')|s_i, a_i, s_j, a_j) \epsilon_\phi^\pi(s_i', s_j') + \\
& \frac{1}{1-\gamma} \sum_{(s_i', s_j') \sim \mathcal{D}'} \mu_\pi((s_i', s_j')|s_i, a_i, s_j, a_j) \epsilon_\phi^\pi(s_i', s_j')
\end{aligned} \tag{34}$$

Recall that $\epsilon_\phi^\pi(s_i, s_j) = \Delta_\phi^\pi(s_i, s_j) - \gamma \mathbb{E}_{\substack{s_i' \sim T_{s_i}^\pi \\ s_j' \sim T_{s_j}^\pi}} [\Delta_\phi^\pi(s_i', s_j')]$, and there exists at least one $G(s_i, s_j)$, such that $(s_i, s_j) \in \mathcal{D}'$. Since the sets $\mathcal{D}$ and $\mathcal{D}'$ are distinct, it follows that there exists a measurement $G$ such that $\epsilon_\phi^\pi(s_i, s_j) = 0$ for all $(s_i, s_j) \in \mathcal{D}$, but $\frac{1}{1-\gamma} \sum_{(s_i', s_j') \sim \mathcal{D}'} \mu_\pi(s_i', s_j'|s_i, a_i, s_j, a_j) \epsilon_\phi^\pi(s_i', s_j') = C$. $\square$

### D.4 Lemma 5

**Lemma 5.** *For any $\tau \in [0, 1)$, $\mathcal{F}_\tau^\pi$ is a $\gamma_\tau$-contraction, where $\gamma_\tau = 1 - 2\alpha(1-\gamma)\min\{\tau, 1-\tau\}$.*

*Proof.* Note that $\mathcal{F}_{1/2}^\pi$ is the standard bisimulation operator for $\pi$, of which the fixed point is $G_\sim^\pi$. To keep the notation succinct, we will replace $G^\pi$ with $G$. For any $G_1$, $G_2$,

$$
\begin{aligned}
&\mathcal{F}_{1/2}^\pi G_1(s_i, s_j) - \mathcal{F}_{1/2}^\pi G_2(s_i, s_j) \\
&= (G_1(s_i, s_j) + \alpha\mathbb{E}^\pi[\delta_i]) - (G_2(s_i, s_j) + \alpha\mathbb{E}^\pi[\delta_j]) \\
&= (1-\alpha)(G_1(s_i, s_j) - G_2(s_i, s_j)) + \alpha\mathbb{E}_\pi[(1-\gamma)|r_{s_i}^\pi - r_{s_j}^\pi| + \gamma G_1(s_i', s_j') \\
&\qquad\qquad\qquad\qquad\qquad\qquad\qquad - (1-\gamma)|r_{s_i}^\pi - r_{s_j}^\pi| - \gamma G_2(s_i', s_j')] \\
&= (1-\alpha)(G_1(s_i, s_j) - G_2(s_i, s_j)) + \alpha\mathbb{E}_\pi[\gamma G_1(s_i', s_j') - \gamma G_2(s_i', s_j')] \\
&\leq (1-\alpha)\|G_1 - G_2\|_\infty + \alpha\gamma\|G_1 - G_2\|_\infty \\
&= (1 - \alpha(1-\gamma))\|G_1 - G_2\|_\infty.
\end{aligned}
\tag{35}
$$

When $\tau \neq \frac{1}{2}$, we introduce two more operators to simplify the analysis:

$$
\begin{aligned}
(\mathcal{F}_+^\pi G_1)(s_i, s_j) &= G(s_i, s_j) + \mathbb{E}^\pi[\delta]_+ \\
(\mathcal{F}_-^\pi G_2)(s_i, s_j) &= G(s_i, s_j) + \mathbb{E}^\pi[\delta]_-
\end{aligned}
\tag{36}
$$

Now we show that both operators meet the Banach-fixed point theorem (e.g. $\|\mathcal{F}_+^\pi G_1 - \mathcal{F}_+^\pi G_2\|_\infty \leq \|G_1 - G_2\|_\infty$). For any $G_1$, $G_2$:

$$
\begin{aligned}
&(\mathcal{F}_+^\pi G_1)(s_i, s_j) - (\mathcal{F}_+^\pi G_2)(s_i, s_j) \\
&= G_1 - G_2 + \mathbb{E}^\pi[[\delta_i]_+ - [\delta_j]_+] \\
&= \mathbb{E}^\pi[G_1 + [\delta_i]_+ - (G_2 + [\delta_j]_+)]
\end{aligned}
\tag{37}
$$

The relationship between $G_1 + [\delta_i]_+$ and $G_2 + [\delta_j]_+$ exists in four cases:

- $\delta_i \geq 0, \delta_j \geq 0$, then

$$
G_1 + [\delta_i]_+ - (G_2 + [\delta_j]_+) = \gamma(G_1(s_i', s_j') - G_2(s_i', s_j')).
\tag{38}
$$

- $\delta_i < 0, \delta_j < 0$, then

$$
G_1 + [\delta_i]_+ - (G_2 + [\delta_j]_+) = G_1(s_i, s_j) - G_2(s_i, s_j).
\tag{39}
$$

- $\delta_i \geq 0, \delta_j < 0$, then

$$
\begin{aligned}
&G_1 + [\delta_i]_+ - (G_2 + [\delta_j]_+) \\
&= (1-\gamma)|r_{s_i}^\pi - r_{s_j}^\pi| + \gamma G_1(s_i', s_j') - G_2(s_i, s_j) \\
&< (1-\gamma)|r_{s_i}^\pi - r_{s_j}^\pi| + \gamma G_1(s_i', s_j') - ((1-\gamma)|r_{s_i}^\pi - r_{s_j}^\pi| + \gamma G_2(s_i', s_j')) \\
&= \gamma(G_1(s_i', s_j') - G_2(s_i', s_j')),
\end{aligned}
\tag{40}
$$

  where the inequality comes from $G_2(s_i, s_j) > (1-\gamma)|r_{s_i}^\pi - r_{s_j}^\pi| + \gamma G_2(s_i', s_j')$.

- $\delta_i < 0, \delta_j \geq 0$, then

$$
\begin{aligned}
&G_1 + [\delta_i]_+ - (G_2 + [\delta_j]_+) \\
&= G_1(s_i, s_j) - ((1-\gamma)|r_{s_i}^\pi - r_{s_j}^\pi| + G_2(s_i', s_j')) \\
&\leq G_1(s_i, s_j) - G_2(s_i, s_j),
\end{aligned}
\tag{41}
$$

  where the inequality comes from $G_2(s_i, s_j) \leq (1-\gamma)|r_{s_i}^\pi - r_{s_j}^\pi| + \gamma G_2(s_i', s_j')$.

As a result, we have $(\mathcal{F}_+^\pi G_1)(s_i, s_j) - (\mathcal{F}_+^\pi G_2)(s_i, s_j) \le \|G_1 - G_2\|_\infty$. Combine $\mathcal{F}_+^\pi$ and $\mathcal{F}_-^\pi$, we can rewrite $\mathcal{F}_\tau^\pi$ as:

$$\begin{aligned}
\mathcal{F}_\tau^\pi G(s_i, s_j) &= G(s_i, s_j) + 2\alpha \mathbb{E}^\pi[\tau[\delta]_+ + (1-\tau)[\delta]_-] \\
&= (1-2\alpha)G(s_i, s_j) + 2\alpha\tau(G(s_i, s_j) + \mathbb{E}^\pi[[\delta]_+] + 2\alpha(1-\tau)(G(s_i, s_j) + \mathbb{E}^\pi[[\delta]_-]) \\
&= (1-2\alpha)G(s_i, s_j) + 2\alpha\tau(\mathcal{F}_+^\pi G_1)(s_i, s_j) + 2\alpha(1-\tau)(\mathcal{F}_-^\pi G_1)(s_i, s_j).
\end{aligned}$$
(42)

What's more

$$\begin{aligned}
\mathcal{F}_{\frac{1}{2}}^\pi G(s_i, s_j) &= G(s_i, s_j) + \alpha\mathbb{E}^\pi[\delta] \\
&= G(s_i, s_j) + \alpha((\mathcal{F}_+^\pi G_1)(s_i, s_j) + (\mathcal{F}_-^\pi G_1)(s_i, s_j) - 2\alpha G(s_i, s_j)) \\
&= (1-2\alpha)G(s_i, s_j) + \alpha((\mathcal{F}_+^\pi G_1)(s_i, s_j) + (\mathcal{F}_-^\pi G_1)(s_i, s_j)).
\end{aligned}$$
(43)

When $\tau > \frac{1}{2}$, for any $G_1$ and $G_2$:

$$\begin{aligned}
&(\mathcal{F}_\tau^\pi G_1)(s_i, s_j) - (\mathcal{F}_\tau^\pi G_2)(s_i, s_j) \\
&= (1-2\alpha)(G_1(s_i, s_j) - G_2(s_i, s_j)) + 2\alpha\tau((\mathcal{F}_+^\pi G_1)(s_i, s_j) - (\mathcal{F}_+^\pi G_2)(s_i, s_j)) \\
&\qquad\qquad\qquad + 2\alpha(1-\tau)((\mathcal{F}_-^\pi G_1)(s_i, s_j) - (\mathcal{F}_-^\pi G_2)(s_i, s_j)) \\
&= (1-2\alpha - 2(1-2\alpha)(1-\tau))(G_1(s_i, s_j) - G_2(s_i, s_j))_2(1-\tau)((\mathcal{F}_{\frac{1}{2}}^\pi G_1)(s_i, s_j) - (\mathcal{F}_{\frac{1}{2}}^\pi G_2)(s_i, s_j)) \\
&\qquad\qquad\qquad - 2\alpha(1-2\tau)((\mathcal{F}_+^\pi G_1)(s_i, s_j) - (\mathcal{F}_+^\pi G_2)(s_i, s_j)) \\
&\le (1-2\alpha - 2(1-2\alpha)(1-\tau))\|G_1(s_i, s_j) - G_2(s_i, s_j)\|_\infty \\
&\qquad\qquad\qquad + 2(1-\tau)(1-\alpha(1-\gamma))\|G_1(s_i, s_j) - G_2(s_i, s_j)\|_\infty \\
&\qquad\qquad\qquad - 2\alpha(1-2\tau)\|G_1(s_i, s_j) - G_2(s_i, s_j)\|_\infty \\
&= (1-2\alpha(1-\tau)(1-\gamma))\|G_1(s_i, s_j) - G_2(s_i, s_j)\|_\infty
\end{aligned}$$
(44)

When $\tau < \frac{1}{2}$, for any $G_1$ and $G_2$:

$$\begin{aligned}
&(\mathcal{F}_\tau^\pi G_1)(s_i, s_j) - (\mathcal{F}_\tau^\pi G_2)(s_i, s_j) \\
&= (1-2\alpha)(G_1(s_i, s_j) - G_2(s_i, s_j)) + 2\alpha\tau((\mathcal{F}_+^\pi G_1)(s_i, s_j) - (\mathcal{F}_+^\pi G_2)(s_i, s_j)) \\
&\qquad\qquad\qquad + 2\alpha(1-\tau)((\mathcal{F}_-^\pi G_1)(s_i, s_j) - (\mathcal{F}_-^\pi G_2)(s_i, s_j)) \\
&= (1-2\alpha - 2\tau(1-2\alpha))(G_1(s_i, s_j) - G_2(s_i, s_j)) + 2\tau((\mathcal{F}_{\frac{1}{2}}^\pi G_1)(s_i, s_j) - (\mathcal{F}_{\frac{1}{2}}^\pi G_2)(s_i, s_j)) \\
&\qquad\qquad\qquad + 2\alpha(1-2\tau)((\mathcal{F}_-^\pi G_1)(s_i, s_j) - (\mathcal{F}_-^\pi G_2)(s_i, s_j)) \\
&\le (1-2\alpha - 2\tau(1-2\alpha))\|G_1(s_i, s_j) - G_2(s_i, s_j)\|_\infty \\
&\qquad\qquad\qquad + 2\tau(1-\alpha(1-\gamma))\|G_1(s_i, s_j) - G_2(s_i, s_j)\|_\infty \\
&\qquad\qquad\qquad + 2\alpha(1-2\tau)\|G_1(s_i, s_j) - G_2(s_i, s_j)\|_\infty \\
&= (1-2\alpha\tau(1-\gamma))\|G_1(s_i, s_j) - G_2(s_i, s_j)\|_\infty.
\end{aligned}$$
(45)

$\square$

## D.5   Lemma 6

**Lemma 6.** *For any $\tau, \tau' \in [0, 1)$ with $\tau' \ge \tau$, and for all $s_i, s_j \in \mathcal{S}$ and any $\alpha$, we have $G_{\tau'} \ge G_\tau$.*

*Proof.* We denote $G_{\tau'}$ is the fixed point of applying the operator $\mathcal{F}_{\tau'}^\pi$, and $G_\tau$ is the fixed point of applying the operator $\mathcal{F}_\tau$. Based on Equation 6, we have:

$$\begin{aligned}
&\mathcal{F}_{\tau'}^\pi G(s_i, s_j) - \mathcal{F}_\tau^\pi G(s_i, s_j) \\
&= (1-2\alpha)G(s_i, s_j) + 2\alpha\tau'\mathcal{F}_+^\pi G(s_i, s_j) + 2\alpha(1-\tau')\mathcal{F}_-^\pi G(s_i, s_j) \\
&\qquad\qquad - ((1-2\alpha)G(s_i, s_j) + 2\alpha\tau\mathcal{F}_+^\pi G(s_i, s_j) + 2\alpha(1-\tau)\mathcal{F}_-^\pi G(s_i, s_j)) \quad (46) \\
&= 2\alpha(\tau' - \tau)(\mathcal{F}_+^\pi G(s_i, s_j) - \mathcal{F}_-^\pi G(s_i, s_j)) \\
&= 2\alpha(\tau' - \tau)\mathbb{E}^\pi[[\delta]_+ - [\delta]_-] \ge 0.
\end{aligned}$$

Therefore $G_{\tau'} > G_\tau$.
$\square$

## D.6 Theorem 7

**Theorem 7.** *In deterministic MDP and fixed finite dataset, we have:*

$$\lim_{\tau \to 1} G_\tau(s_i, s_j) = \max_{\substack{a_i \in \mathcal{A}, a_j \in \mathcal{A} \\ s.t. \ \pi_\beta(a_i|s_i)>0, \pi_\beta(a_j|s_j)>0}} G_\sim^*((s_i, a_i), (s_j, a_j)). \tag{47}$$

*where $G_\sim^*((s_i, a_i), (s_j, a_j))$ is a fixed-point measurement constrained to the dataset and defined on state-action space $\mathcal{S} \times \mathcal{A}$ as*

$$G_\sim^*((s_i, a_i), (s_j, a_j)) = |r(s_i, a_i) - r(s_j, a_j)| + \gamma \mathbb{E}_{\substack{s_i' \sim T_{s_i}^{\pi_\beta} \\ s_j' \sim T_{s_j}^{\pi_\beta}}} \left[ \max_{\substack{a_i' \in \mathcal{A}, a_j' \in \mathcal{A} \\ s.t. \ \pi_\beta(a_i'|s_i')>0, \pi_\beta(a_j'|s_j')>0}} G_\sim^*((s_i', a_i'), (s_j', a_j')) \right]. \tag{48}$$

*Proof.* First, we can easily proof that $G_\sim^*(s_i, a_i, s_j, a_j)$ is a fixed point. Define the corresponding operator of $G_\sim^*$ is $F^*$, we can know that $F^*$ is a contraction. Then, we have

**Corollary 16.** *For any $\tau$, $s_i, s_j \in \mathcal{S}$ we have*

$$G_\tau(s_i, s_j) \le \max_{\substack{a_i \in \mathcal{A}, a_j \in \mathcal{A} \\ s.t. \ \pi_\beta(a_i|s_i)>0, \pi_\beta(a_j|s_j)>0}} G_\sim^*((s_i, a_i), (s_j, a_j)) \tag{49}$$

*Proof.* The proof follows from the observation that convex combination is smaller than maximum. $\square$

Besides, we also have

**Lemma 17.** *Let $X$ be a real-valued random variable with a bounded support and supremum of the support is $x^*$. Then,*

$$\lim_{\tau \to 1} m_\tau = x^*$$

*Proof.* Same as the Lemma 1 in [28]. One can show that expectiles of a random variable have the same supremum $x^*$. Moreover, for all $\tau_1$ and $\tau_2$ such that $\tau_1 < \tau_2$, we get $m_{\tau_1} \le m_{\tau_2}$. Therefore, the limit follows from the properties of bounded monotonically non-decreasing functions. $\square$

Combining Corollary 16 and Lemma 17, we can obtain the above.

$\square$

## D.7 Theorem 8

**Theorem 8.** *(Value bound based on on-policy bisimulation measurements in terms of encoder error). Given an MDP $\widetilde{\mathcal{M}}$ constructed by aggregating states in an $\omega$-neighborhood, and an encoder $\phi$ that maps from states in the original MDP $\mathcal{M}$ to these clusters, the value functions for the two MDPs are bounded as*

$$\left| V^\pi(s_i) - \widetilde{V}^\pi(\phi(s_i)) \right| \le \frac{2\omega + \hat{\Delta}}{c_r(1 - \gamma)}. \tag{50}$$

*where $\hat{\Delta} := \|\hat{G}_\sim^\pi - \hat{G}_\phi^\pi\|_\infty$ is the approximation error.*

*Proof.* Let the reward function be bounded as $R \in [0, 1]$, $\phi : \mathcal{S} \to \widetilde{\mathcal{S}}$, and $\phi(s_i) = \phi(s_j) \Rightarrow \hat{G}_\phi^\pi(s_i, s_j) = |\phi(s_i) - \phi(s_j)| \le 2\omega$, we can conduct an aggregat MDP $\widetilde{M} = (\widetilde{\mathcal{S}}, \mathcal{A}, \widetilde{T}, \widetilde{R})$. Let $\xi$ be

a measure on $\mathcal{S}$. Following Lemma 8 in [27], we have that:

$$\left| V^\pi(s_i) - \widetilde{V}^\pi(\phi(s_i)) \right| \leq \frac{c_r^{-1}}{\xi(\phi(s))} \int\limits_{z \in \phi(s)} c_R \left| r^\pi(s) - r^\pi(z) \right|$$

$$+ (1-\gamma) \left| \int\limits_{s' \in \mathcal{S}} (T^\pi(s'|s) - T^\pi(s'|z)) \frac{c_r \gamma}{1-\gamma} V^\pi(s') ds' \right| d\xi(z) + \gamma \|V^\pi - \widetilde{V}^\pi\|_\infty$$

$$\leq \frac{c_r^{-1}}{\xi(\phi(s))} \int\limits_{z \in \phi(s)} G_\sim^\pi(s,z) d\xi(z) + \gamma \|V^\pi - \widetilde{V}^\pi\|_\infty \tag{51}$$

Thus, taking the supremum on the LHS, we have:

$$(1-\gamma) \left| V^\pi(s_i) - \widetilde{V}^\pi(\phi(s_i)) \right| \leq \frac{c_r^{-1}}{\xi(\phi(s))} \int\limits_{z \in \phi(s)} G_\sim^\pi(s,z) d\xi(z)$$

$$\leq \frac{c_r^{-1}}{\xi(\phi(s))} \int\limits_{z \in \phi(s)} \hat{G}_\phi^\pi(s,z) + \|G_\sim^\pi - \hat{G}_\phi^\pi\|_\infty d\xi(z) \tag{52}$$

$$= \frac{c_r^{-1}}{\xi(\phi(s))} \int\limits_{z \in \phi(s)} (2\omega + \hat{\Delta}) d\xi(z)$$

$$= c_r^{-1}(2\omega + \hat{\Delta}).$$

Therefore,

$$\left| V^\pi(s_i) - \widetilde{V}^\pi(\phi(s_i)) \right| \leq \frac{2\omega + \hat{\Delta}}{c_r(1-\gamma)}, \tag{53}$$

$\square$

# E    Understanding of Theorem 7

**Theorem 7.** *In deterministic MDP and fixed finite dataset, we have:*

$$\lim_{\tau \to 1} G_\tau(s_i, s_j) = \max_{\substack{a_i \in \mathcal{A}, a_j \in \mathcal{A} \\ \text{s.t. } \pi_\beta(a_i|s_i) > 0, \pi_\beta(a_j|s_j) > 0}} G_\sim^*((s_i, a_i), (s_j, a_j)). \tag{54}$$

*where $G_\sim^*((s_i, a_i), (s_j, a_j))$ is a fixed-point measurement constrained to the dataset and defined on the state-action space $\mathcal{S} \times \mathcal{A}$ as*

$$G_\sim^*((s_i, a_i), (s_j, a_j)) = |r(s_i, a_i) - r(s_j, a_j)| + \gamma \mathbb{E}_{\substack{s_i' \sim T_{s_i}^{\pi_\beta} \\ s_j' \sim T_{s_j}^{\pi_\beta}}} \left[ \max_{\substack{a_i' \in \mathcal{A}, a_j' \in \mathcal{A} \\ \text{s.t. } \pi_\beta(a_i'|s_i') > 0, \pi_\beta(a_j'|s_j') > 0}} G_\sim^*((s_i', a_i'), (s_j', a_j')) \right]. \tag{55}$$

Given the MDP specified by the tuple $(\mathcal{S}, \mathcal{A}, T, R)$, we construct a lifted MDP $(\widetilde{\mathcal{S}}, \widetilde{\mathcal{A}}, \widetilde{T}, \widetilde{R})$, by taking the state space to be $\widetilde{\mathcal{S}} = \mathcal{S}^2$, the action space to be $\widetilde{\mathcal{A}} = \mathcal{A}^2$, the transition dynamics to be given by $\widetilde{T}_{\tilde{s}}^{\tilde{a}}(\tilde{s}') = \widetilde{T}_{(s_i, s_j)}^{(a_i, a_j)}((s_i', s_j')) = T_{s_i}^{a_i}(s_i') T_{s_j}^{a_j}(s_j')$ for all $(s_i, s_j), (s_i', s_j') \in \mathcal{S}^2$, $a_i, a_j \in \mathcal{A}$, and the action-independent rewards to be $\widetilde{R}_{\tilde{s}} = \widetilde{R}_{(s_i, s_j)} = |r_{s_i}^\pi - r_{s_j}^\pi|$ for all $s_i, s_j \in \mathcal{S}$. The Bellman evaluation operator $\widetilde{\mathcal{F}}^{\tilde{\pi}}$ for this lifted MDP at discount rate $\gamma$ under the policy $\tilde{\pi}(\tilde{a}|\tilde{s}) = \tilde{\pi}(a_i, a_j|s_i, s_j) = \pi(a_i|s_i)\pi(a_j|s_j)$ is given by (for all $G^\pi \in \mathbb{R}^{\mathcal{S} \times \mathcal{S}}$ and $(s_i, s_j) \in \mathcal{S} \times \mathcal{S}$):

$$(\widetilde{\mathcal{F}}^{\tilde{\pi}} Q^*)(\tilde{s}, \tilde{a}) = \widetilde{R}_{\tilde{s}, \tilde{a}} + \gamma \sum_{\tilde{s} \in \widetilde{\mathcal{S}}} \widetilde{T}_{\tilde{s}}^{\tilde{a}}(\tilde{s}') \max_{\tilde{a} \in \widetilde{\mathcal{A}}} Q^*(\tilde{s}', \tilde{a}'). \tag{56}$$

Though similar, Equation 55 has more constraints as it requires the possibility of $\pi_\beta(a_i'|s_i')$ and $\pi_\beta(a_j'|s_j')$ are larger than zero in the dataset. As such, we may also change the Equation 56 to:

$$(\widetilde{\mathcal{F}}^{\tilde{\pi}} Q^*)(\tilde{s}, \tilde{a}) = \widetilde{R}_{\tilde{s},\tilde{a}} + \gamma \sum_{\tilde{s} \in \widetilde{\mathcal{S}}} \widetilde{T}_{\tilde{s}}^{\tilde{a}}(\tilde{s}') \max_{\substack{\tilde{a}' \in \mathcal{A} \\ \text{s.t. } \tilde{\pi}_\beta(\tilde{a}'|\tilde{s}') > 0}} Q^*(\tilde{s}', \tilde{a}'). \tag{57}$$

This is, indeed, equivalent to the *in-sample*-style Q function in [28]. Intuitively, $G_\sim^*((s_i, a_i), (s_j, a_j))$ can be interpreted as the optimal state-action value function $Q^*(\tilde{s}, \tilde{a})$ in a lifted MDP $\widetilde{M}$. Then $G_\sim^\pi((s_i, a_i), (s_j, a_j))$ is the state-action value function $Q^\pi(\tilde{s}, \tilde{a})$ that associated with policy $\pi$, and $G_\sim(s_i, s_j)$ as a state value function $V(\tilde{s})$. And therefore, we can connect our expectile-based bisimulation operator to the lifted MDP, where we can use the conventional analytics tools in RL to analyze bisimulation operators.

# F    Additional Experiments

## F.1    Ablation Study - Value of Expectile

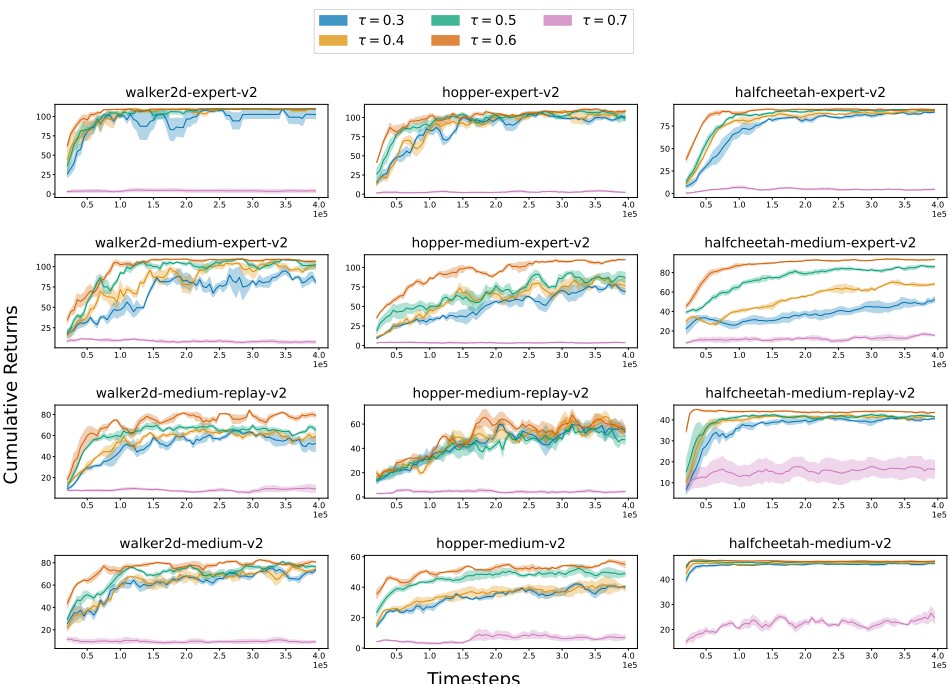

Figure 6: Performance comparison on 12 D4RL tasks over 10 seeds with one standard error shaded in the default setting.

Here we present the ablation study of setting different expectile $\tau \in \{0.3, 0.4, \cdots, 0.7\}$ in Figure 6 to investigate the effect of the critical hyper-parameter in EBS. The experimental results demonstrate that the final performance gradually improves with a larger $\tau$. Notably, the most superior performance is achieved when $\tau$ equals 0.6. However, when $\tau$ further increases to 0.7, the agent's performance suffers a sharp decline. We hypothesize that this could be due to the value function possibly exploding when $\tau$ is set to larger values, subsequently leading to poorer performance outcomes. This is as expected since the over-large $\tau$ leads to the overestimation error caused by neural networks. The experimental results demonstrate that we can balance a trade-off between minimizing the expected bisimulation residual and evaluating "optimal" measurement solely on the dataset by choosing a suitable $\tau$.

## F.2    Ablation Study - Effectiveness of Reward Scaling

In the experiment, we set $\gamma$ as 0.99 and $c_r$ will be $1 - \gamma = 0.01$ accordingly in **RS**. In this ablation experiment, we considered different combinations of min-max normalization/standardization and

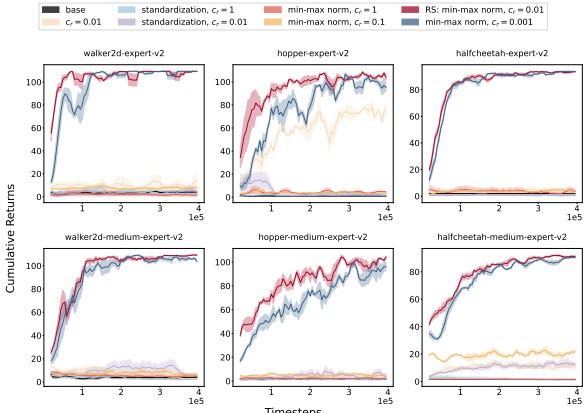

Figure 7: Ablation studies on 6 D4RL tasks over 3 seeds with one standard error shaded.

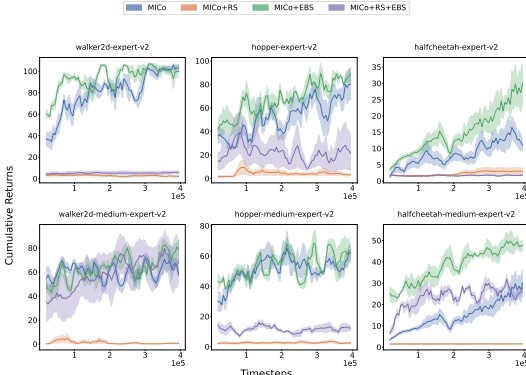

Figure 8: Ablation studies on 6 D4RL tasks over 3 seeds on MICo.

various value of $c_r$ (including 1, 0.1, 0.01, and 0.001). The results in Figure 7 are consistent with our analysis in Section 5.2. The last two show better gains. As **RS** has tighter bounds, it excels in most datasets, validating our theory.

### F.3 Case Study on MICo

As we illustrate in Section 5.2 and Section 6, Results in Figure 8 show that an unsuitable reward scaling dramatically decreases the performance while applying EBS will increase the performance in many datasets.

## G Additional Related Works

Here we present a brief introduction of all the baselines we used in the experiments:

**TD3BC [16]** add a behavior cloning term to regularize the policy of the TD3 [18] algorithm, achieves a state-of-the-art performance in Offline settings.

**DrQ+BC [35]** combining data augmentation techniques with the TD3+BC method, which applies TD3 in the offline setting with a regularizing behavioral-cloning term to the policy loss. The policy objective is: $\pi = \underset{\pi}{\mathrm{argmax}} \mathbb{E}_{(s,a)\sim\mathcal{D}} \left[ \lambda Q(s, \pi(s)) - (\pi(s) - a)^2 \right]$

**DRIML [38] and HOMER [39]** (Time Contrastive methods) learn representations which can discriminate between adjacent observations in a rollout and pairs of random observations.

**CURL [31]**  (Augmentation Contrastive method) learns a representation that is invariant to a class of data augmentations while being different across random example pairs.

**Inverse Model [44]**  (One-Step Inverse Models) predict the action taken conditioned on the previous and resulting observations.

# H    Additional Discussion

## H.1    The severity of the proposed problem

**How do bisimulation-based objectives perform in other (online or goal-conditioned) settings?**
Various methods, such as DBC [54], MICo [6], SimSR [52], and PSE [1], have consistently demonstrated positive results in online settings, regardless of the presence of distractors. This evidence supports the efficacy of bisimulation techniques in online settings. Additionally, GCB [23] excelled in goal-conditioned environments, ExTra [46] showcased the power of bisimulation metric in exploration, and HiP-BMDP [55] successfully incorporated bisimulation into multi-task settings, highlighting its superior performance, all mostly in online settings too, with little work in offline RL. These studies suggest that when tailored to specific environments, bisimulation methods can excel. Despite these works, bisimulation methods have had little success when extended to offline settings, and our motivation is to tackle this problem.

**While bisimulation objectives used in the offline setting are directly affected by missing transitions, many other representation objectives may not.**    When referring to state representation learning, using bisimulation in offline settings presents challenges due to the two issues we outlined: the presence of missing transitions and inappropriate reward scales. Concurrently, there exists other representation objectives, like CURL [31], ATC [49], which focus on pairs of states without the explicit necessity for transition information. As a consequence, they do not explicitly require accounting for missing transitions or reward scaling in their objectives. This absence of direct influence sets them apart from bisimulation-based methods. Yet, we consider that bisimulation-based techniques have a theoretical edge and have proven effective in online settings, Thus, we deem that our work is impactful in that it delivers a proof that bisimulation can be successful offline.

**Compounded effect for bisimulation principle in offline settings.**    In online scenarios, state representations and policies are updated concurrently, while in offline settings, state representation is pre-trained before policy learning, with the two phases completely decoupled. Errors during representation learning in offline settings can have a compounded effect on policy learning, leading to significant issues. This is the reason that missing transitions is particularly harmful to the bisimulation principle in offline settings. Although reward scales affect bisimulation universally, as offline settings require pretraining state embedding, any major discrepancy between this fixed representation and the policy parameter space can further undermine the learning process. Hence, the proposed solutions hold promise for enhancing bisimulation's efficiency in offline settings.

## H.2    Suitability of different techniques

**EBS**    We provide EBS as a general method, which is applicable to all bisimulation-based objectives, given that they all adhere to the foundational principle of bisimulation. This principle revolves around the contraction mapping properties similar to the value iteration. Whenever there's an intent to employ bisimulation in offline scenarios, with an aim to reduce the Bellman residual for approximating the fixed point, the outlined challenge emerges. Consequently, EBS holds the potential to enhance any bisimulation-based method, regardless of the distance they use.

**RS**    In essence, the given theoretical analysis is applicable across all bisimulation-based objectives. However, the precise settings for hinge on the foundational distance. For instance, SimSR uses the cosine distance which has definitive bounds. As a result, we need to infer the ideal setting from Equation 10 and Theorem 8. In contrast, the MICo-like distance and DBC employ L-K distance and L1 distance respectively, having bounds ranging from . Consequently, they can adapt to more value settings. We propose our approach as a general method/principle to employ a novel bisimulation metric or distance, especially in the context of offline RL.

Table 3: The exact values of bisimulation error and bisimulation bellman residual

| Transition number | 100 | 500 | 1000 | 2000 |
|---|---|---|---|---|
| Bisimulation error | 0.2792 | 0.2891 | 0.2880 | 0.2915 |
| Bisimulation Bellman residual | 0.003 | 0.0009 | 0.0032 | 0.0016 |

# I  Empirical estimation of bisimulation error

In this section, we would like to conduct a toy experiment to empirically show that the bisimulation error could possibly be larger than bisimulation bellman residual in fixed/finite datasets.

**Data collection**    To collect the evaluation dataset, we utilize TD3 [18] (a deterministic algorithm) instead of SAC [22] to avoid stochasticity. Firstly, we train a TD3 agent using the rlkit [45] codebase until convergence. Then, we collect 10k transitions and select specific transitions (such as 10, 100, 1000, 5000...) from these 10k transitions with uniform probability to form the evaluation dataset $\mathcal{D}$. For determining termination, we follow the settings described in [18] and [22], considering a state terminal only if termination occurs before 1000 timesteps. If termination occurs before 1000 timesteps, we set $\gamma = 0$; otherwise, we set $\gamma = 0.99$.

**Computation**    Given an evaluation dataset $\mathcal{D}$, the bisimulation Bellman residual $\epsilon_\phi$ is computed by $\frac{1}{|\mathcal{D}|} \sum_{((s_i, ai), (s_j, a_j)) \sim \mathcal{D}} (G_\phi(s_i, s_j) - (| r(s_i, a_i) - r(s_j, a_j) + \gamma G_\phi(s_i, s_j)))^2$, and the bisimulation error $\Delta_\phi$ is computed by $\frac{1}{\mathcal{D}} \sum_{(s_i, s_j) \sim \mathcal{D}} (G_\phi(s_i, s_j) - G_\sim(s_i, s_j))^2$, where $G_\sim$ denotes the corresponding fixed point measurement. Since directly computing $G_\sim$ is challenging, we compute $|V^\pi(s_i) - V^\pi(s_j)|$ instead, as they should be equal when considering the measurement is the fixed point and the transition is deterministic. To compute $V^\pi(s_i)$, we reset the MuJoCo environment to the specific state $s_i$ and ran the policy for 1000 timesteps. Since the environment and policy are deterministic, a single trajectory is sufficient to estimate the true value.

The results are presented in Table 3, which indicates that the bisimulation error on finite datasets is indeed larger than the bisimulation bellman residual.

