# OpenReview forum: "Understanding and Addressing the Pitfalls of Bisimulation-based Representations in Offline Reinforcement Learning"
_NeurIPS.cc/2023/Conference — NeurIPS 2023 poster_

### Official Review · Reviewer_BVFy · 2023-06-22

**Soundness:** 3 good
**Presentation:** 4 excellent
**Contribution:** 3 good
**Rating:** 6
**Confidence:** 2

**Summary:**

Bisimulation methods tend to fail when applied to offline RL. The authors performed a bisimulation error analysis and concluded that missing transitions, inevitable when training with a fixed offline dataset, lead to inaccurate metric predictions. Therefore, training policies based on the representation learned with inaccurate predictions will perform poorly. The paper proposes a $\tau$-expectile-based bisimulation operator to overcome this challenge. This operator achieves fully in-sample learning in the limit $\tau\to 1$. The authors also note that reward scaling is crucial when the underlying state-space metric is bounded. Experiments conducted on state-based and pixel-based offline RL problems show that both expectile-based bisimulation operator and reward scaling have a significant impact on the final quality of the learned representation.

**Strengths:**

The authors mathematically analyzed the properties of the original bisimulation operator and the novel expectile-based bisimulation operator. The mathematical techniques used in the proofs and the careful analysis of the error bounds will be very helpful to other researchers trying to work on applying bisimulation-metric-based representation learning to offline RL. The expectile-based based bisimulation operator the authors proposed is also interesting and effective under various settings, as the experimental results reveal. Finally, the paper is overall well-written and easy to understand.

**Weaknesses:**

It is doubtful that learning a bisimulation operator in the offline setting generally fails because of the reasons the paper presents. To facilitate learning by diversifying trajectories, a lot of RL environments impose a limit on the episode length. An environment with a time limit of $T$ should have states that an agent can arrive within $T+1$ timesteps but not in $T$ timesteps. Those states satisfy the conditions of Proposition 4, which means learning a bisimulation operator will fail. This is not the case.

Also, according to line 238, unlike the original bisimulation operator, learning the expectile-based bisimulation operator will succeed because we only consider state pairs with corresponding actions in the dataset. Suppose this is the reason why learning an expectile-based bisimulation operator succeeds. Then why not just omit the last and the second-last states of each trajectory in the dataset during training while using the original bisimulation operator? Then the next-state pairs of every state-pairs left will have corresponding actions in the dataset. It is hard to imagine such a simple solution will work.

**Questions:**

Is the proposed algorithm also effective in the online RL setting? Also, it would be helpful for the readers if the paper introduces the definition of $G^\pi$ and a *lifted MDP* before mentioning them in equation (2) and line 139, respectively.

**Limitations:**

The authors have adequately addressed the limitations of their work, and I do not believe this work will have a potential negative societal impact.

---

> ### Author Rebuttal · Authors · 2023-08-09
>
> Thank you for your detailed review and feedback on our paper. We have included new experimental results and some general discussion, please refer to our generic response and uploaded file to address these points.
>
> - **Question: It is doubtful that learning a bisimulation operator in the offline setting generally fails because of the reasons the paper presents. To facilitate learning by diversifying trajectories, a lot of RL environments impose a limit on the episode length. An environment with a time limit of T should have stated that an agent can arrive within T+1 timesteps but not in T timesteps. Those states satisfy the conditions of Proposition 4, which means learning a bisimulation operator will fail. This is not the case.**
>
> - **Response**: While it is true that many RL environments truncate the episode length (for example, some tasks in Mujoco may have a maximum length of 500), this chiefly pertains to pragmatic considerations surrounding sample efficiency. Primarily, within the Preliminaries, we emphasized that our framework is predicated on an infinite horizon, thus necessitating the discount factor gamma (in contrast, the finite horizon scenario does not require the gamma factor, a distinction that sets it apart from our problem formulation). Secondly, the predicament articulated in Proposition 4 is unaffected by the length of trajectories or the truncation of episodes. At its core, whenever the bisimulation Bellman residual is employed as a surrogate objective in an offline setting, this dilemma surfaces. This arises from the reality that as long as the dataset remains incomplete, the relationship between $\epsilon_{\phi}^{\pi}$ and $\Delta^{\pi}_{\phi}$ does not strictly hold.
>
> - **Question: According to line 238, unlike the original bisimulation operator, learning the expectile-based bisimulation operator will succeed because we only consider state pairs with corresponding actions in the dataset. Suppose this is the reason why learning an expectile-based bisimulation operator succeeds. Then why not just omit the last and the second-last states of each trajectory in the dataset during training while using the original bisimulation operator? Then the next-state pairs of every state-pairs left will have corresponding actions in the dataset. It is hard to imagine such a simple solution will work.**
>
> - **Response**: We may not have entirely understood the reviewer's question. Even if we exclude the last and second-to-last states, the third-to-last states remain. For these particular states, we must still compute $\hat{\epsilon}$, necessitating the sampling of subsequent states. Since we have already dispensed with the second-to-last states, this calculation falls outside the confines of the dataset.
>
> - **Question: Is the proposed algorithm also effective in the online RL setting? Also, it would be helpful for the readers if the paper introduces the definition of G^\pi  and a lifted MDP before mentioning them in equation (2) and line 139, respectively.**
>
> - **Response**: Theoretically, as long as the dataset is incomplete, this issue will arise. Specifically, in on-policy settings, the expected bisimulation residual proves adequate for learning representations (as established in Theorem 3); conversely, in off-policy settings, it serves as a deficient estimator, as the missing transitions disrupts the relationship between $\epsilon_{\phi}^{\pi}$ and $\Delta^{\pi}_{\phi}$ (as outlined in Proposition 4), though it is marginally better than offline settings.
> Thanks for the recommendation, and we will contemplate reorganizing the structure of the relevant paragraph to enhance its readability and coherence.

---

> > ### Comment · Reviewer_BVFy · 2023-08-13
> >
> > Thank you for the rebuttal.
> >
> > **Q1**: I am still not convinced that the incompleteness of the dataset is the primary reason why learning the bisimulation operator fails in the offline RL setting. Bisimulation-based online RL algorithms DBC[1], SimSR[2], and MICO[3] perform well on the DeepMind Control Suite tasks that have a horizon of 1000 timesteps, which is the same as the horizon of D4RL MuJoCo tasks, while assuming infinite horizon by incorporating a discount factor. However, as I previously mentioned, the dataset can never be complete when episodes are truncated at a certain timestep. Then by Proposition 4, DBC, SimSR, and MICO should fail to learn the bisimulation metric and should perform poorly.
> >
> > **Q2**: I misunderstood the meaning of Equation (8) when I was writing the review. Sorry for the confusion caused.
> >
> > &nbsp;
> > ### References
> >
> > [1]: Reference [48] of the paper
> >
> > [2]: Reference [46] of the paper
> >
> > [3]: Reference [5] of the paper

---

> > > ### Author Response · Authors · 2023-08-14
> > > **Response to Reviewer BVFy**
> > >
> > > Thanks for the reply.
> > >
> > > Once again, we would like to emphasize that **in offline settings, the dataset is indeed fixed** and beyond our control regarding how the agent collects it. Therefore, the dataset may suffer from incompleteness, biases, and lack of diversity [A, B]. This "incompleteness" is even more severe compared to online learning settings, where online reinforcement learning involves exploration to interact with the environment and collect data, potentially including new states that have not been seen before.
> > >
> > > With all due respect, it is important to note that **the state incompleteness of the dataset is unrelated to how the episodes are truncated**. In the DMC tasks, almost every state has the opportunity to be reached within the given truncated timesteps of 1000 due to exploration, indicating that the agent in some sense is able to perceive the ``complete’’ state space in online settings.
> > >
> > > In extreme scenarios, there might be cases where only one state (let's call it Y) can reach a final state (let's call it X), while the probability of any other state reaching X is 0, regardless of the action taken. Another extreme case could occur in goal-conditioned settings, where the goal is 1001 steps away from the initial point, and truncating the horizon at 1000 steps would make the goal unreachable. However, it is important to note that these cases are unlikely to occur in practice. Truncation is typically done to prevent trajectories from becoming excessively long and containing less useful information for training, rather than intentionally preventing the agent from reaching its goal. This is supported by the fact that none of the environments in the DMC tasks [C] evaluated in DBC, SimSR, or MICo align with these two cases.
> > >
> > > We hope that this explanation can help the reviewer understand the motivation for proposing our approach. If there are any other confusion or questions that the reviewer may have, please do not hesitate to let us know. We are happy to provide any additional clarification or information that may be needed to ensure a thorough understanding of our work.
> > >
> > > [A]: Sergey Levine, Aviral Kumar, George Tucker, and Justin Fu. Offline reinforcement learning: Tutorial, review, and perspectives on open problems. arXiv preprint arXiv:2005.01643, 2020.
> > >
> > > [B]: Schweighofer K, Hofmarcher M, Dinu M C, et al. Understanding the effects of dataset characteristics on offline reinforcement learning[C]//Deep RL Workshop NeurIPS 2021. 2021.
> > >
> > > [C]: Tassa Y, Doron Y, Muldal A, et al. Deepmind control suite[J]. arXiv preprint arXiv:1801.00690, 2018.

---

> > > > ### Author Response · Authors · 2023-08-15
> > > > **Further clarification**
> > > >
> > > > The main focus of our discussion may revolve around the distinction between **state incompleteness** and **trajectory incompleteness**. In our original paper, we primarily referred to "state incompleteness", where not all states are covered in the dataset. However, we inadvertently overlooked the concept of "trajectory incompleteness",  which relates to the presence of missing transitions within certain trajectories. We understand that this oversight may have caused confusion for the reviewer. We apologize for any misconceptions and assure you that we will revise our descriptions accordingly in the forthcoming revisions.

---

> > > > ### Comment · Reviewer_BVFy · 2023-08-15
> > > >
> > > > Since the dynamics $T$ of the MuJoCo tasks are deterministic and the behavior policy $\pi_\beta$ used to create the D4RL datasets&mdash;at least the single policy ones&mdash;are also deterministic, which makes the bisimulation operator the following:
> > > >
> > > > $$d(s_1, s_2)=c_r|r(s_1, \pi_\beta(s_1))-r(s_2, \pi_\beta(s_2)|+c_kd(T(s_1, \pi_\beta(s_1)), T(s_2, \pi_\beta(s_2))).$$
> > > >
> > > > Doesn't this mean the only "state incompleteness" comes from the end-of-trajectory states?

---

> > > > > ### Author Response · Authors · 2023-08-15
> > > > > **Further response**
> > > > >
> > > > > > the behavior policy used to create the D4RL datasets—at least the single policy ones—are also deterministic
> > > > >
> > > > > We do not make any assumptions about how the agent collects data in offline settings. However, even if we do consider it, the data collection agent is indeed not deterministic but rather stochastic. Please refer to paper[1]: **"The medium dataset is generated by first training a policy online using Soft Actor-Critic (Haarnoja et al., 2018a), early-stopping the training..."**. It means the collection agent is SAC, whose policy is indeed stochastic.
> > > > >
> > > > > > Doesn't this mean the only "state incompleteness" comes from the end-of-trajectory states?
> > > > >
> > > > > For instance, let's consider the state $X=[0,1]$ is the end-of-trajectory state, and we have $s_1=[0,0]$ with $a_1=[0,1]$ in trajectories in order to reach state $X$, e.g., $\tau=(s_0=[0,-1],a_0=[0,1],s_1=[0,0],a_1=[0,1],s_2=X=[0,1],a_2=...)$. However, when we truncate the trajectory as the length of 2, we may exclude the pair $(s_1,a_1, X)$ (or $s_1,a_1,s_2$). Nevertheless, it is still conceivable for another trajectory that includes the pair $s_2=[0,-1]$ with $a_2=[0,2]$ to reach state $X$, e.g., $\tau_2=(s_0'=[0,-1],a_0'=[0,2],s_1'=X=[0,1],a_1'=...)$. That is the difference between "state incompleteness" and "trajectory incompleteness". And this is irrelevant to how bisimulation defines.

---

> > > > > > ### Comment · Reviewer_BVFy · 2023-08-15
> > > > > >
> > > > > > **Q1.** I thought the policy was deterministic because [rlkit](https://github.com/rail-berkeley/rlkit/tree/master) evaluates policy using the mean of the learned Gaussian. However, after rereading the [trajectory generation script](https://github.com/Farama-Foundation/D4RL/blob/71a9549f2091accff93eeff68f1f3ab2c0e0a288/scripts/generation/mujoco/collect_data.py), I admit that it looks like they collected the data using a stochastic policy, although nothing's certain until the authors of D4RL make the policy pickle files publicly available. Anyway, I do think it is okay to assume the behavior policy is deterministic since it is extremely unlikely to visit a certain state more than once.
> > > > > >
> > > > > > **Q2.** Let us assume deterministic dynamics and a deterministic policy and consider two trajectories: $(s_0, a_0, r_0, s_1, a_1, r_1, \dotsc, s_N)$ and $(s_0', a_0', r_0', s_1', a_1', r_1', \dotsc, s_M')$, where $N\le M$. Then
> > > > > >
> > > > > > $$d(s_0, s_0')=|r_0-r_0'|+\gamma d(s_1, s_1')=\dotsb=\sum_{t=0}^{N-1}\gamma^t|r_t-r_t'|+\gamma^N d(s_N, s_N').$$
> > > > > >
> > > > > > If the bisimulation Bellman residual is zero, then the bisimulation error $\Delta(s_0, s_0')$ is $\gamma^N\Delta(s_N, s_N')$. That is, the bisimulation error is propagated from the end-of-trajectory states to others. This is the reason why I kept mentioning trajectory truncation. If $N$ is sufficiently large, the error will be negligible, which means the approximated bisimulation metric between states "far away" from end-of-trajectory states will be quite accurate.

---

> > > > > > > ### Comment · Reviewer_BVFy · 2023-08-15
> > > > > > >
> > > > > > > It feels like our discussion is going in circles, so I would like to clarify my concerns.
> > > > > > >
> > > > > > > 1. By the argument stated in my review, there are states that satisfy the premises of Proposition 4, even in the online RL setting. However, bisimulation-metric-based algorithms show good performance in online RL settings.
> > > > > > > 2. This means the bisimulation error depends on how "incomplete" the dataset is. The "incompleteness" is negligible in the online RL setting, so the bisimulation metric is accurately approximated. Assuming the authors' claim is true, the "incompleteness" is large in the offline RL setting, and the approximation is erroneous.
> > > > > > > 3. My take on this problem is the following:
> > > > > > >     * The approximated bisimulation metrics are probably quite accurate, at least for most of the states in the dataset. (Based on my argument in the previous comment.)
> > > > > > >     * Bisimulation-metric-based offline RL algorithms fail for other reasons, such as being unable to generalize to unseen states or deviating too far away from the behavior policy.
> > > > > > >
> > > > > > > I would gladly change my mind if the authors could provide empirical evidence for their claim.
> > > > > > >
> > > > > > > 1. Devise a metric that can measure the "incompleteness" of the dataset and theoretically analyze the relationship between this metric and the bisimulation error. Estimate the value of this metric on the offline RL datasets and show that they are "too incomplete" for bisimulation-metric-based algorithms to work.
> > > > > > > 2. Or instead, directly estimate the bisimulation error on the offline RL datasets and show that they are large.

---

> > > > > > > > ### Author Response · Authors · 2023-08-16
> > > > > > > > **Response to Reviewer BVFy**
> > > > > > > >
> > > > > > > > > The approximated bisimulation metrics are probably quite accurate, at least for most of the states in the dataset. (Based on my argument in the previous comment.)
> > > > > > > >
> > > > > > > > - Being accurate in some states is not sufficient. This is the difficulty with Offline RL: mistakes / bad estimates cannot be fixed through interaction with the environment as with Online RL. Bisimulation adds another layer of risk in how the states are represented. Proposition 4 formalizes this risk.
> > > > > > > >
> > > > > > > > > Bisimulation-metric-based offline RL algorithms fail for other reasons, such as being unable to generalize to unseen states or deviating too far away from the behavior policy.
> > > > > > > >
> > > > > > > > - We would argue that this is not another reason, but a refinement of it. The distribution shift caused by the deviation from the behavior policy is the source of all troubles. This has been lengthily analyzed in the Offline RL literature. Our analysis exhibits another bisimulation-related potential pitfall that is induced by the dataset's incompleteness of state coverage.
> > > > > > > >
> > > > > > > > > Devise a metric that can measure the "incompleteness" of the dataset and theoretically analyze the relationship between this metric and the bisimulation error. Estimate the value of this metric on the offline RL datasets and show that they are "too incomplete" for bisimulation-metric-based algorithms to work.
> > > > > > > >
> > > > > > > > - The incompleteness of the state coverage is difficult to measure with a single scalar as it depends on many factors.
> > > > > > > >
> > > > > > > > > Or instead, directly estimate the bisimulation error on the offline RL datasets and show that they are large.
> > > > > > > >
> > > > > > > > - Thank you for the suggestion. We have carried out an empirical study to demonstrate that this case indeed occurs in offline settings.
> > > > > > > >
> > > > > > > > - **Data collection**: To collect the evaluation dataset, we utilize TD3 (a deterministic algorithm) instead of SAC to avoid stochasticity. Firstly, we train a TD3 agent using the rlkit[1] codebase until convergence. Then, we collect 10k transitions and select specific transitions (such as 10, 100, 1000, 5000...)  from these 10k transitions with uniform probability to form the evaluation dataset $\mathcal{D}$. For determining termination, we follow the settings described in [2] and [3], considering a state terminal only if termination occurs before 1000 timesteps. If termination occurs before 1000 timesteps, we set $\gamma=0$; otherwise, we set $\gamma=0.99$.
> > > > > > > >
> > > > > > > >
> > > > > > > > - **Computation**: Given an evaluation dataset $\mathcal{D}$, the bisimulation Bellman residual $\epsilon_\phi$ is computed by $\frac{1}{|\mathcal{D}|} \sum_{((s_i,a_i),(s_j,a_j))\sim\mathcal{D}} \big(G_{\phi}(s_i,s_j)-(|r(s_i,a_i)-r(s_j,a_j)+\gamma G_{\phi}(s_i,s_j))\big)^2$ , and the bisimulation error $\Delta_\phi$ is computed by $\frac{1}{\mathcal{D}} \sum_{(s_i,s_j)\sim\mathcal{D}} (G_\phi(s_i,s_j)-G_\sim(s_i,s_j))^2$, where $G_\sim$ denotes the corresponding fixed point measurement. Since directly computing $G_\sim$ is challenging, we compute $|V^\pi(s_i) - V^\pi(s_j)|$ instead, as they should be equal when considering the measurement is the fixed point and the transition is deterministic.  To compute $V^{\pi}(s_i)$, we reset the MuJoCo environment to the specific state $s_i$ and ran the policy for 1000 timesteps. Since the environment and policy are deterministic, a single trajectory is sufficient to estimate the true value. By comparing the exact values of  $\epsilon_\phi$ and $\Delta_\phi$, we can verify our statements. Unfortunately, we are unable to upload any figures on openreview, so we present the result values below:
> > > > > > > >
> > > > > > > > | Transition number             | 100  | 500  | 1000 | 2000 |
> > > > > > > > | ----------------------------- | ---- | ---- | ---- | -----|
> > > > > > > > | Bisimulation error            |0.2792 | 0.2891 | 0.2880 | 0.2915 |
> > > > > > > > | Bisimulation Bellman residual | 0.003  | 0.0009 | 0.0032 | 0.0016 |
> > > > > > > >
> > > > > > > > - The results clearly demonstrate that there exist cases where the bisimulation error exceeds the bisimulation Bellman residual, which aligns with Proposition 4 in our manuscript. Besides, we would like to draw the reviewer's attention to a paper [4] that presents a similar result to our bisimulation, but in off-policy settings where the severity is somewhat weakened. In the upcoming revision, we will incorporate these results into our work. We sincerely appreciate the reviewer's valuable feedback, which has greatly contributed to enhancing the overall quality of our paper.
> > > > > > > >
> > > > > > > >
> > > > > > > >
> > > > > > > > [1]: rlkit: A simple Reinforcement Learning library
> > > > > > > >
> > > > > > > > [2]: Fujimoto S, Hoof H, Meger D. Addressing function approximation error in actor-critic methods. ICML 2018: 1587-1596.
> > > > > > > >
> > > > > > > > [3]: Haarnoja T, Zhou A, Abbeel P, et al. Soft actor-critic: Off-policy maximum entropy deep reinforcement learning with a stochastic actor. ICML 2018: 1861-1870.
> > > > > > > >
> > > > > > > > [4]: Scott Fujimoto, David Meger, Doina Precup, Ofir Nachum, Shixiang Shane Gu: Why Should I Trust You, Bellman? The Bellman Error is a Poor Replacement for Value Error. ICML 2022: 6918-6943

---

> > > > > > > > > ### Comment · Reviewer_BVFy · 2023-08-18
> > > > > > > > >
> > > > > > > > > Thank you for sharing the experiment results. I have raised my score to 6 but lowered my confidence.

---

> > > > > > > > > > ### Author Response · Authors · 2023-08-20
> > > > > > > > > > **Thank you for updating the scores**
> > > > > > > > > >
> > > > > > > > > > Thank you very much for updating the scores for the paper. We sincerely appreciate your in-depth discussion with us, which engaged in helping us defend our paper. We will make corresponding revisions and updates based on your feedback in the upcoming revision.
> > > > > > > > > >
> > > > > > > > > > If you have any further feedback on the draft, please let us know too.

---

### Official Review · Reviewer_ndNc · 2023-07-06

**Soundness:** 3 good
**Presentation:** 4 excellent
**Contribution:** 4 excellent
**Rating:** 7
**Confidence:** 3

**Summary:**

The paper investigates why bisimulation-based methods, which are known to work well in the online setting, fail in the offline case. It hypothesizes that missing transitions and reward scaling issues are a source of this failure, and proposes a way to mitigate these issues. It includes experiments on various benchmarking tasks.

**Strengths:**

1. According to me, this is an important problem to solve. Recently bisimulation-based representation learning methods have become popular but have focused mostly on the online setting. The insights from this paper are illuminating, especially Theorem 3 and Proposition 4.
2. I like that they considered the MICO distance since unlike other metrics, MICO is a diffuse metric and has additional complications due to the non-zero self-distances.


**Weaknesses:**

It appears that the approach is limited to a single and known behavior policy. I wonder how this could be used in the multiple and unknown behavior policy setting, which is starting to become a more popular setting in offline RL.

**Questions:**

1. I’d point authors to this new paper by the MICO authors: https://openreview.net/forum?id=nHfPXl1ly7 to see how their approach fits in with this kernel-based bisimulation approach.
2. How do reward normalization schemes impact the results? For example, subtracting min and dividing by max-min (where min/max are the min/max rewards in the dataset), or 0/1 mean/std normalization etc? Are these beneficial when computing the bisimulation metric?
3. Why is the Effective Dimension a good metric in the offline RL case (Figure 1)? The paper just plots the results without justifying the usage. I ask this by considering other offline RL metrics for representations such as distribution shift and coverage [1, 2] in terms of the state-action feature representations.
4. From the experiments, it appears that MICO and SimSR were applied as-is. I wonder if it would be useful to apply an off-policy version of say MICO by either sampling from the policy getting trained or using some version of importance sampling?
5. It would be interesting to include some preliminary results in the appendix on off-policy evaluation (not just control) (keep the main results in tact; I am referring more to an exploratory analysis for the appendix).
6. How is maximization and subject-to constraints (eqn 8) satisfied in continuous action setting?

[1] Instabilities of Offline RL with Pre-Trained Neural Representation. Wang et al.

[2] Learning Bellman Complete Representations for Offline Policy Evaluation. Chang et al.


**Limitations:**

See above.

N/A societal impact.

---

> ### Author Rebuttal · Authors · 2023-08-09
>
> Thank you for your detailed review and feedback on our paper. We have included new experimental results and some general discussion, please refer to our generic response and uploaded file to address these points.
>
> - **Question: It appears that the approach is limited to a single and known behavior policy. I wonder how this could be used in the multiple and unknown behavior policy setting, which is starting to become a more popular setting in offline RL.**
>
> - **Response**: First, we want to clarify that we do not have access to the behavior policy that was used to collect data. Second, a recent paper [A] proves under mild conditions, that the dataset’s transitions distribution (aka the occupancy measure) obtained from multiple behavior policies can be replicated with a single Markovian behavior policy. This theoretical result rigorously allows any Offline RL algorithm to pretend that the dataset has been generated with the equivalent single Markovian behavior policy and thus the multiplicity (and non-stationarity) of the behavior policy does not present any additional challenge to the Offline RL task.
>
> [A] Laroche, R. and Tachet Des Combes, R. On the Occupancy Measure of Non-Markovian Policies in Continuous MDPs. ICML, 2023.
>
> - **Question: I’d point authors to this new paper by the MICO authors to see how their approach fits in with this kernel-based bisimulation approach.**
>
> - **Response**: Thanks the reviewer for bringing this work to our attention. In the general response, we delve into the suitability of the two techniques we've proposed. The kernel-based bisimulation approach, dubbed KSMe, which adheres to the bisimulation principle, could inherently reap benefits from the EBS technique when applied in offline scenarios. Given that $supp(\mathcal{R})\in[-1,1]$ and $k_1(x, y)=1-\frac{1}{2}\left|r_x^\pi-r_y^\pi\right|$, the immediate similarity is constrained within the interval $[0,1]$. Importantly, the kernel isn't subjected to a specific bound, thus nullifying the need to contemplate the RS technique (aligning with the relationship between KSMe and MICo as elucidated in the said paper - an issue that MICo bypasses as well). Furthermore, it preemptively processes the scale of the rewards prior to considering the bisimulation operator, thus mirroring an insight our paper provides, which is, the significance of a well-scaled reward.
>
> - **Question: How do reward normalization schemes impact the results? For example, subtracting min and dividing by max-min (where min/max are the min/max rewards in the dataset), or 0/1 mean/std normalization etc? Are these beneficial when computing the bisimulation metric?**
>
> - **Response**: This is a good question. To delve deeper into the relevance of the reward scaling technique, we have incorporated an ablation study. In a nutshell, the 0/1 mean/std normalization (referred to as "standardization" in the attached document) does not deliver favorable empirical outcomes. We have furnished the corresponding findings in Figure 1 of the uploaded pdf file and a detailed account in the general response.
>
> - **Question: Why is the Effective Dimension a good metric in the offline RL case (Figure 1)? The paper just plots the results without justifying the usage. I ask this by considering other offline RL metrics for representations such as distribution shift and coverage [1, 2] in terms of the state-action feature representations.**
>
> - **Response**: Thanks for bringing these measurements to our attention. Indeed, the foundational concepts guiding these methodologies bear striking resemblance, as all emanate from feature reachability or the rank covariance of the feature. They are unified in their aspiration to gauge the efficacy of representations, and our choice of one over the other for evaluation was largely discretionary.
>
> - **Question: From the experiments, it appears that MICO and SimSR were applied as-is. I wonder if it would be useful to apply an off-policy version of say MICO by either sampling from the policy getting trained or using some version of importance sampling? It would be interesting to include some preliminary results in the appendix on off-policy evaluation**
>
> - **Response**: Thank you for this insightful suggestion! The question of how importance sampling techniques (such as per-decision importance sampling) fare with MICo and SimSR indeed makes for an engaging question. Theoretically, using importance sampling as an illustration, it can be implemented for off-policy value estimation. However, the instability of importance sampling for long horizons is a recognized challenge. Given that bisimulation measurements bear relevance to the value function in some regard, we anticipate that this approach will grapple with analogous challenges within the bisimulation principle. It's worth noting, though, that despite conventional importance sampling's potential limitations, recent works in off-policy evaluation have emerged to mitigate these concerns. This constitutes a promising avenue of research, and we may well explore the integration and adaptation of these advancements within the context of bisimulation in offline scenarios. We intend to contemplate including a discussion and analysis of related works in the appendix. Once more, we extend our sincere gratitude to the reviewer for these invaluable recommendations.
>
> - **Question: How is maximization and subject-to-constraints (eqn 8) satisfied in continuous action setting?**
>
> - **Response**: The maximization operation in Equation 8 only occurs when $\tau=1$, and it represents solely a theoretical determination. In practical application, we in fact employ Equation 6, and as a result, the maximization operation does not participate in concrete implementations.

---

> > ### Comment · Reviewer_ndNc · 2023-08-12
> >
> > Thank you to the authors for answering my questions. I will keep the score as-is. This seems like a useful paper, especially given the advent of many bisimulation-based representation learning methods and recent work that has shown they tend to fail in the offline setting [1].
> >
> > [1]: Mengjiao Yang, Ofir Nachum. Representation Matters: Offline Pretraining for Sequential Decision Making. ICML 2021

---

> > > ### Author Response · Authors · 2023-08-15
> > > **Response to Reviewer ndNc**
> > >
> > > Thank you for the positive feedback. We appreciate your efforts in reviewing our work. We will reflect your suggestions in the final version to enable it to be a high-quality paper.

---

### Official Review · Reviewer_KC23 · 2023-07-12

**Soundness:** 3 good
**Presentation:** 3 good
**Contribution:** 3 good
**Rating:** 6
**Confidence:** 4

**Summary:**

The paper discusses the pitfall of bisimulation-based methods in offline RL. It identifies missing transitions in the finite dataset as a significant problem for bisimulation-based methods, leading to ineffective estimation. The paper also highlights the importance of reward scaling in bounded cosine distance method.

To address these issues, the paper proposes using the expectile operator for the measurement learning in offline RL, as expectile operator helps prevent overfitting to incomplete data. Additionally, the paper analyzes the upper bound of metric w.r.t. the reward scale and proposes a reward scaling strategy to reduce representation collapse.

**Strengths:**

This paper establishes a connection between the Bellman and bisimulation operators to understand the gap for bisimulation-based approaches in online and offline RL settings. It defines the bisimulation Bellman residual and the bisimulation error and analyzes the gap of between minimizing them.

This paper also analyzes the bounds of the bimiluation-based metric and how the reward scaling affects the cosine distance. It demonstrates that setting the reward coefficient to $1-\gamma$ improves performance compared to using SimSR.

**Weaknesses:**

The motivation for utilizing the expectile operator could be explained more clearly. While expectile regression is applicable to both online [1] and offline RL, its relevance to addressing Proposition 4 and preventing overfitting in the offline RL setting requires further elaboration.

The authors propose reward scaling as a solution to mitigate the limitation imposed by the cosine distance, which has a range of [0, 2]. This range imposes an upper bound on the value of $c_r$. However, Theorem 8 suggests that a larger value of $c_r$ leads to a more accurate approximation of the value function. An alternative approach to overcome the limitations of the cosine distance would be to use other distance metrics such as L1 or L2 distance. Hence, it would be beneficial for the authors to clarify the rationale behind selecting cosine distance over these alternatives.

In the comparison experiments between SimSR+RS(+EBS) and MICo(+EBS), it is possible that RS improves performance due to the numerical stability of neural networks. By upper bounding the range of $G$ to [0, 1], which can be stably predicted by neural networks, RS might contribute to the observed performance enhancement.

The bisimulation Bellman residual $\epsilon_{\phi}^{\pi}$ may not be minimized to zero if $G_{\phi}^{\pi}$ is a cosine distance. Consider the case of $s_i = s_j$. $G_{\phi}^{\pi}(s_i, s_j) = 0$ because it is cosine distance. However, $FG_{\phi}^{\pi}(s_i, s_j)$ is a Łukaszyk–Karmowski distance because  $\gamma  E_{s^{\prime}_i, s^{\prime}_j} G_\phi^\pi(s^{\prime}_i, s^{\prime}_j)$

is a Łukaszyk–Karmowski distance as $s^{\prime}_i$ and $s^{\prime}_j$ are sampled [2],

 and $|r_i^{\pi} - r_j^{\pi}|$ is untractable and usually alternated to $E_{r_i, r_j}|r_i - r_j|$, which is again a Łukaszyk–Karmowski distance[3]. Therefore, $FG_{\phi}^{\pi}(s_i, s_j) \geq G_{\phi}^{\pi}(s_i, s_j)$ if $s_i = s_j$, the equality is taken only if $G_{\phi}^{\pi}(s^{\prime}_i, s^{\prime}_j) = 0$ for all $s^{\prime}_i$ and $s^{\prime}_j$ and

$r_i = r_j$ for all $r_i$ and $r_j$. This can destroy the Proposition 4 because not all $\epsilon_{\phi}^{\pi}$ is zero. However, MICo distance does not meet this issue.

------
Reference

[1] Rowland et al. Statistics and Samples in Distributional Reinforcement Learning. ICML 2019.

[2] Castro et al. MICo: Improved representations via sampling-based state similarity for Markov decision processes. NeurIPS 2021.

[3] Chen et al. Learning Representations via a Robust Behavioral Metric for Deep Reinforcement Learning. NeurIPS 2022.

**Questions:**

1. Lemma 5 and 6 require a citation to [4].
1. How does the value of $c_r$ affect the value bound according to Theorem 8 and Theorem 15?
1. How is Eq. (29) derived from Eq. (28) in Theorem 15 in the Supplementary Material?
1. It is confusing that the main text defines $\Delta$ and $\epsilon$ as absolute values, while the supplementary material defines them as differences. The authors should use different notations (rather than the same $\Delta$ and $\epsilon$) in the supplementary material.
1. What is the value of $\tau$ in Table 1?
1. Some implementation details are missing, including the learning rate and network architecture.

-----
Reference

[4] Ma et al. Offline Reinforcement Learning with Value-based Episodic Memory. ICLR 2022.

**Limitations:**

The authors should clarify the motivation of expectile operator and refine the theoretical analysis with cosine distance.

---

> ### Author Rebuttal · Authors · 2023-08-09
>
> Thank you for your detailed review and feedback on our paper. We have included new experimental results and some general discussion, please refer to our generic response and uploaded file to address these points.
>
> - **Question: While expectile regression is applicable to both online [1] and offline RL, its relevance to addressing Proposition 4 and preventing overfitting in the offline RL setting requires further elaboration.**
>
> - **Response**: An intuitive explanation is that when considering the on-policy settings, i.e., bisimulation Bellman residuals and bisimulation error are both coupled with policy $\pi$, thereby Theorem 3 can be derived. This is to say that if we can access the optimal policy $\pi^*$ and we apply bisimulation on it, the problem mentioned in Proposition 4 naturally does not exist. However, in practice, it is actually impossible to obtain bisimulation corresponding to the optimal when the dataset is fixed. Therefore, we can only use expectile to strike a balance between behavior and optimal, to alleviate this problem.
>
> - **Question: It would be beneficial for the authors to clarify the rationale behind selecting cosine distance over these alternatives.**
>
> - **Response**: We wish to gently draw the reviewer's attention to the fact that the selection of cosine distance was specified in SimSR[1] instead of ours. The rationale behind their selection likely stems from cosine distance having zero self-distance and being suitable for constructing a unit hypersphere. Notably, in contrast, our proposed methodology does not require a specific distance. In our experiments, we also applied EBS on MICo and showed its superiority. For a more comprehensive explanation of the suitability of techniques we have presented, please refer to the general response.
>
> - **Question: In the comparison experiments between SimSR+RS(+EBS) and MICo(+EBS), it is possible that RS improves performance due to the numerical stability of neural networks.**
>
> - **Response**: To verify the effectiveness of RS, we additionally construct an ablation study, which consists of several different combinations to reward scaling strategy. Please refer to Figure 1 in our uploaded pdf file and our general response for detailed explanations.
>
> - **Question: The bisimulation Bellman residual may not be minimized to zero if $G^\pi_\phi$ is a cosine distance, This can destroy the Proposition 4 because not all $\epsilon^\pi_\phi$  is zero. However, MICo distance does not meet this issue.**
>
> - **Response**: We perceive the possibility that the reviewer might postulate that the MICo distance is impervious to the issue elucidated in Proposition 4. It's important to note, however, that Proposition 4 doesn't purport that the bisimulation Bellman residual will invariably minimize to zero. Rather, it describes a scenario wherein even with the bisimulation Bellman residual at its nadir - zero, there might yet be pairs of states manifesting a bisimulation error that is non-zero. This scenario is entirely unconnected with the distance upon which bisimulation objectives are contingent.
>
> - **Question: Lemma 5 and 6 require a citation to [4].**
>
> - **Response**: Thanks for the suggestion! We will add the citations in the revision.
>
> - **Question: How does the value of c_r affect the value bound according to Theorem 8 and Theorem 15?**
>
> - **Response**: In relation to Theorem 8, as $c_r$ inhabits the denominator of the right-hand side of Equation 11, an enlargement in the magnitude of $c_r$ will precipitate a diminution in the value boundary. Conversely, in Theorem 15, given that $c_r$ solely influences the reward's scale without contributing to Equation 27's derivation, we remain capable of formulating an equation akin to Equation 30. The sole alteration would be the incorporation of the coefficient $c_r$ within the confines of the expectation.
>
> - **Question: How is Eq. (29) derived from Eq. (28) in Theorem 15 in the Supplementary Material?**
>
> - **Response**: Assuming we interpret $\Delta_{\phi}^{\tilde{\pi}}(\tilde{x})$ as a value function and $\epsilon_{\phi}^{\tilde{\pi}}(\tilde{x})$ as reward, then Equation 28 can be re-written as $V(s)=\sum\gamma \mathbb{E}_{s'\sim P^{\pi}}[r(s')]=\frac{1}{1-\gamma}\mathbb{E}_{s'\sim P^{\pi}}[r(s')]$.  Consequently, Equation 29 can be deduced in concordance with this formulation.
>
> - **Question: It is confusing that the main text defines $\Delta$ and $\epsilon$ as absolute values, while the supplementary material defines them as differences. The authors should use different notations (rather than the same $\Delta$ and $\epsilon$) in the supplementary material.**
>
> - **Response**: Our motivation stems from the pursuit of preserving succinctness and autonomy in the primary manuscript's notation. Nonetheless, during the appendix's proofing process, we encounter stages necessitating the consideration of forms devoid of absolute values. We will contemplate amending the symbols within the appendix to enhance lucidity when revisions to the manuscript become feasible.
>
> - **Question: What is the value of tau in Table 1?**
>
> - **Response**: The value of $\tau$ is 0.6 for SimSR, and $\tau$ is 0.7 for MICo for all datasets in V-D4RL datasets, we will add this in revision.
>
> - **Question: Some implementation details are missing, including the learning rate and network architecture.**
>
> - **Response**: For d4rl tasks, we employ an Adam optimizer with learning rate of 3e-4, and an identical encoder, i.e., a 4-layer MLP activated by ReLU, succeeded by another linear layer. For v-d4rl tasks, we employ an Adam optimizer with learning rate of 1e-4, and an identical encoder, i.e., four convolutional layers activated by ReLU, succeeded by another linear layer. We will include the corresponding descriptions in revision.

---

> > ### Comment · Reviewer_KC23 · 2023-08-20
> > **Reply to Authors**
> >
> > Thanks for your response. Most of my concerns have been addressed. I have raised the score to 6.

---

> > > ### Author Response · Authors · 2023-08-20
> > > **Response to Reviewer KC23**
> > >
> > > Thank you for raising your score. We appreciate your efforts in reviewing our work. We will reflect your suggestions in the final version to enable it to be a high-quality paper.

---

### Official Review · Reviewer_LXid · 2023-07-15

**Soundness:** 3 good
**Presentation:** 2 fair
**Contribution:** 3 good
**Rating:** 6
**Confidence:** 3

**Summary:**

This paper focuses on understanding and ameliorating the relatively poor performance of bisimulation-based representations in offline RL. The authors identify two main issues: overfitting to incomplete data and reward scaling. When the offline dataset is missing states, the representation that’s learned may not actually reflect the true bisimulation metric of the underlying MDP even if the associated Bellman residual over the dataset is zero. Their approach is to regularize learning with expectile regression to avoid overfitting to the incomplete data. With regards to reward scaling, the authors show that a larger reward scale can produce a more accurate approximation of the value function. Experiments are carried out on continuous control tasks with both proprioceptive and visual inputs.

**Strengths:**

- The theoretical analysis seems carefully done and gives useful, practical insights, and the connection of the bisimulation update operator to policy evaluation is interesting.

- The careful evaluation of the proposed approach in the D4RL experiments is fairly convincing that the modifications proposed by the authors produce some improvement in performance.

**Weaknesses:**

- In the D4RL results, SimSR + RS + EBS only clearly outperforms the other approaches on 3, possibly 4, of the 12 settings.

- It would be helpful to have either more qualitative analysis of the existing experiments or a toy experiment to provide intuition as to the resulting behavioral differences and success/failure cases induced by the authors’ approach rather than just performance plots.

- Running the V-D4RL benchmark over only 3 seeds seems rather low, and no measures of variance are provided. This stands in contrast to the D4RL results, for which not only full learning curves with error shading are shown, but also IQM measurements.

- More discussion (possibly in the appendix) regarding the variation in performance across different datasets would be helpful.

- Clarity could be improved in some areas, particularly in providing more background to the introduction of the expectile-based operator how it aids in in-sample learning / prevents overfitting to incomplete data. More precise language would also be helpful in the reward scaling section.


**Questions:**

- Can the authors provide learning curves + IQM analysis for the V-D4RL experiment?

- Could the authors run experiments with MICo + RS (and MICo + RS + EBS) as well? If RS is not applicable to MICo for some reason, could the authors add a note either in the appendix or main text explaining why?

**Limitations:**

The authors do an adequate job acknowledging the limitations of their approach, but perhaps more discussion would be helpful in the appendix. I don’t see any direct negative societal implications of this work.

---

> ### Author Rebuttal · Authors · 2023-08-09
>
> Thank you for your detailed review and feedback on our paper. We have included new experimental results and some general discussion, please refer to our generic response and uploaded file to address these points.
>
> - **Question: In the D4RL results, SimSR + RS + EBS only clearly outperforms the other approaches on 3, possibly 4, of the 12 settings.**
>
> - **Response**: Our proposed methods RS and EBS are essential for SimSR to achieve success on all datasets. By incorporating RS+EBS, the agent pretrained with SimSR demonstrates superior performance. Additionally, EBS significantly improves MICo's performance on 7 out of 12 datasets, providing strong evidence supporting our analysis. Considering that TD3+BC is already recognized for its rapid convergence and superior performance, it is expected that enhancements may not be substantial on simple datasets. In RL, sample efficiency and convergence rate are crucial. Faster convergence holds more practical significance and is preferred by the community, even if two algorithms have similar performance. This is particularly important due to the extensive training iterations required in RL. Our methodology becomes even more evident in datasets where TD3+BC struggles to converge. We observe that our method converges more rapidly on 8 out of 12 datasets, as evident from the performance curve.
>
> - **Question: It would be helpful to have either more qualitative analysis of the existing experiments or a toy experiment to provide intuition rather than performance plots.**
>
> - **Response**: We have provided examples for each case in Lines 180-188 and Figure 1 respectively.  To illustrate the failure case of missing transitions, we consider a scenario where only (s_i, a_i, r, s′_i) and (s_j, a_j, r, s′_j) are present in the dataset, with both rewards equal to zero. The bisimulation Bellman residual could be zero, but the bisimulation error could still be positive in this case. Failure cases like this can be found in many instances. To highlight the importance of selecting a suitable reward scale, we analyze the upper bound of the cosine-based bisimulation - SimSR. We also demonstrate that without a proper scale of reward, the effective dimension of the state feature will dramatically decrease, as shown in Figure 1 Left. These examples aim to provide an intuitive understanding of our proposed techniques. It is possible that the reviewer did not notice these descriptions due to the way we presented. We will consider adjusting the structure to make them more prominent.
>
>
> - **Question: Running the V-D4RL benchmark over only 3 seeds seems rather low, and no measures of variance are provided. This stands in contrast to the D4RL results, for which not only full learning curves with error shading are shown, but also IQM measurements. Providing learning curves + IQM analysis for the V-D4RL experiment.**
>
> - **Response**: Thanks, we provided a more comprehensive performance comparison on the V-D4RL benchmark in the uploaded file, which is averaged over 10 seeds with one standard error shaded. We have also included IQM measurements among these datasets to demonstrate their effectiveness. These results can be found in Figure 3 and Figure 4 in the attached PDF file. We believe these additional experiments will help address concerns regarding the reliability of our method.
>
> - **Question: More discussion (possibly in the appendix) regarding the variation in performance across different datasets would be helpful.**
>
> - **Response**: Thank you for your suggestion. We will add these in revisions. Different datasets inherently possess unique characteristics, leading to variations in performance [A]. All 12 tasks in Figure 2 are considered somewhat relevant, with each column representing a specific task domain and each row representing the form of data collection. Due to variations in data quality and task nature, it is normal for algorithms to exhibit performance differences across these datasets. More details can be found in [B].
>
> [A]: Schweighofer K, Hofmarcher M, Dinu M C, et al. Understanding the effects of dataset characteristics on offline reinforcement learning[C]//Deep RL Workshop NeurIPS 2021. 2021.
>
> [B]: Fu J, Kumar A, Nachum O, et al. D4rl: Datasets for deep data-driven reinforcement learning[J]. arXiv preprint arXiv:2004.07219, 2020.
>
> - **Question: Clarity could be improved in some areas, particularly in providing more background to the introduction of the expectile-based operator how it aids in in-sample learning / prevents overfitting to incomplete data. More precise language would also be helpful in the reward scaling section.**
>
> - **Response**: Thanks, we will provide more relevant background information and descriptions to help readers follow the technical details more easily in revisions.
>
> - **Question: Could the authors run experiments with MICo + RS (and MICo + RS + EBS) as well? If RS is not applicable to MICo for some reason, could the authors add a note either in the appendix or main text explaining why?**
>
> - **Response**: We have additionally added the corresponding results in Figure 2 in uploaded pdf file, the detailed explanation can be found in general response. We also provided a discussion of the suitability of our proposed techniques in general response. We hope these results/discussions can satisfy reviewer’s concerns.

---

> > ### Comment · Reviewer_LXid · 2023-08-14
> > **Response**
> >
> > Thank you for your response! I found the additional details and experiments helpful, and am more confident about acceptance. I will increase my score.

---

> > > ### Author Response · Authors · 2023-08-15
> > > **Response to Reviewer LXid**
> > >
> > > Thank you for raising your score. We appreciate your efforts in reviewing our work. We will reflect your suggestions in the final version to enable it to be a high-quality paper.

---

### Official Review · Reviewer_aN21 · 2023-07-22

**Soundness:** 4 excellent
**Presentation:** 3 good
**Contribution:** 3 good
**Rating:** 6
**Confidence:** 2

**Summary:**

This paper addresses the problem of bisimulation-based representations in offline reinforcement learning tasks, which are less effective than alternative methods. To address this issue, the paper proposes a tailored reward scaling strategy and an auxiliary loss function based on bisimulation metrics to learn robust state representations. The experiments show that the proposed method outperforms existing methods on a range of locomotion tasks in the DeepMind Control Suite. Ablation studies suggest that the proposed reward scaling strategy and bisimulation-based loss function are critical to the performance of the method. Overall, I believe this work makes a valuable contribution to the field of reinforcement learning by improving the effectiveness of bisimulation-based representations in offline RL tasks.

**Strengths:**

**Motivation and intuition**
- The motivation for addressing the pitfalls of bisimulation-based representations in offline reinforcement learning is convincing. The authors clearly explain the challenges faced by bisimulation-based approaches in offline RL tasks and the need for solutions to improve their performance. Their analysis reveals that missing transitions in the dataset, which often occur in offline settings, are particularly harmful to the bisimulation principle

**Novelty**
- The idea of utilizing expectile operators to prevent overfitting to incomplete data is intuitive and convincing, which is theoretically supported by the paper's analysis. This paper presents an effective way to implement this idea, which has not been explored in previous works.

**Technical contribution**
- The proposed solutions, an expectile-based operator and a tailored reward scaling strategy, for addressing the pitfalls of bisimulation-based representations in offline RL tasks seem compelling, especially when dealing with limited datasets. The authors provide a thorough analysis of the performance gains achieved by their proposed solutions in various benchmark suites.

**Clarity**
- The overall writing is clear and well-organized. The authors utilize figures well to illustrate the ideas, and Figure 1 clearly shows the results with and without reward scaling.
- The paper gives clear descriptions in both theoretical and intuitive ways. The notations, formulations, and theorems are well-explained in the appendix, making it easy for readers to follow the technical details.

**Experimental results**
- The experimental results are impressive and demonstrate the effectiveness of the proposed framework in improving the performance of bisimulation-based approaches in offline RL tasks. The authors provide clear visualizations of the results, and Figure 3 particularly provides a clear comparison of the performance gains achieved by the proposed framework over the two state-of-the-art bisimulation-based algorithms, MICo and SimSR.

**Reproducibility**
- The code is provided, which helps understand the details of the proposed framework.
- Given the clear description in the main paper and the details provided in the appendix, I believe reproducing the results is possible.

**Weaknesses:**

**Ablation study**
The proposed framework comprises two components: an expectile-based operator and a tailored reward scaling strategy. Conducting an ablation study to verify the proposed methods to some naive ones would be good. Take the tailored reward scaling strategy for example, one possible naive reward scaling strategy could be to multiply the reward signal by a constant factor, such as 0.1 or 10, and observe the effect on the performance of the bisimulation-based RL algorithm. Another possible method could be to clip the reward signal to a certain range, such as [-1, 1], and observe the effect on the performance.

**Limitation**
I notice that the authors say that Since MICo does not necessitate a particular upper bound, RS may harm its performance, leading them to exclude the MICo+RS results from the main experiments. It would be good to elaborate more on when the two methods, an expectile-based operator and a tailored reward scaling strategy, can help improve the performance of given algorithms.

**Experiment setup**
 I think that adding manipulation and navigation environments to the experimental evaluation of the proposed method would be a valuable addition to the paper. While the locomotion environments used in the current evaluation provide a good testbed for evaluating the proposed method, it would be beneficial to evaluate the method in a broader range of environments to better understand its generalizability and effectiveness across different types of tasks instead of only evaluating on locomotion environments.

**Questions:**

See above

**Limitations:**

See above

---

> ### Author Rebuttal · Authors · 2023-08-09
>
> Thank you for your review and favorable assessment of our work! We appreciate the time and effort you have put into evaluating our work. Based on your feedback, we included additional rebuttal experiments, on benchmark D4RL tasks and on visual image-based offline tasks, to demonstrate the significance of our approach (please see attached rebuttal pdf). We also prefer a more in-depth discussion regarding the aptness of the techniques we have proposed. Please refer to our generic response for all reviewers to address these points. We proceed with our response to each concern as follows:
>
> - **Question: Conducting an ablation study to verify the proposed methods to some naive ones would be good. Take the tailored reward scaling strategy for example, one possible naive reward scaling strategy could be to multiply the reward signal by a constant factor…**
>
> - **Response**: Thanks for your suggestion!  We concur that an extensive ablation analysis would indeed enrich our work significantly. We have incorporated an ablation study of the reward scaling strategy in the attached PDF file. In brief, as deliberated in Lines 260-270 of our paper, only when we employ min-max normalization and set $c_r\in[0,1-\gamma]$, can we guarantee alignment and attain commendable performance; the other combinations of reward scaling strategy invariably fall short across all datasets. Please refer to the general response for more details.
>
> - **Question: It would be good to elaborate more on when the two methods, an expectile-based operator and a tailored reward scaling strategy, can help improve the performance of given algorithms.**
>
> - **Response**: Thanks for your insightful suggestion. We have provided a general response discussing the suitability of different techniques, and we will also add the corresponding descriptions in the revision. Briefly, the theoretical analysis of both techniques is suitable for all bisimulation-based objectives, but their practical applications may vary. Specifically, the Expectile-Based Scaling (EBS) can be applied to any bisimulation objectives when we consider using the bisimulation principle in offline settings. As for Reward Scaling (RS), we only need to consider it when the distance used in bisimulation is tightly bounded. For example, the range of L1/L2 distance and MICo-like distance (diffuse metric) is $[0,+\infty]$, the range of cosine distance is $[-1,1]$. Therefore, bisimulation with L1/L2/MICo, which fall into the former category, do not need to use RS; bisimulation with cosine distance or some similar distance, which fall into the latter category, do require RS. We hope this clarifies any confusion, and we are happy to provide further explanation if needed.
>
> - **Question: adding manipulation and navigation environments to the experimental evaluation of the proposed method would be a valuable addition to the paper.**
>
> - **Response**: We fully agree with your suggestion that including experiments in manipulation and navigation environments would indeed strengthen the generalizability and robustness of our research. However, given the limited time window of the current rebuttal stage and our computational resources, we have prioritized completing some other experimental supplements first. For instance, we have focused on conducting the ablation study on reward scaling in d4rl tasks and running experiments with more seeds in the vd4rl tasks. Regarding the navigation tasks and other experiments,  accomplishing them necessitates more time. We plan to explore these experiments later and conduct an evaluation on larger-scale datasets to provide more robust evidence of the potential of the bisimulation method.
>
> Once again, we appreciate the insightful comments and are grateful for the opportunity to improve the quality of our work. If you have any further suggestions or guidance, we would be more than happy to incorporate them into our work.

---

> > ### Comment · Reviewer_aN21 · 2023-08-10
> > **Thanks for the rebuttal**
> >
> > I appreciate the author's rebuttal, which addresses some of my concerns/confusion. I believe this work studies a promising problem and provides meaningful insights. Yet, I still hope to see a more comprehensive set of experiments, including robot arm manipulation (such as D4RL kitchen) and navigation. In sum, I am still slightly leaning toward accepting this paper, while I won't fight for this paper if the majority of the reviewers have a different opinion.

---

> > > ### Author Response · Authors · 2023-08-15
> > > **Response to Reviewer aN21**
> > >
> > > Thank you for the positive feedback. We appreciate your efforts in reviewing our work. We will reflect your suggestions in the final version to enable it to be a high-quality paper.

---

### Official Review · Reviewer_vGKe · 2023-07-27

**Soundness:** 2 fair
**Presentation:** 2 fair
**Contribution:** 2 fair
**Rating:** 4
**Confidence:** 3

**Summary:**

Summary
-------

This submission\'s goal is to understand why bisimulation methods
suffers in the offline RL setting. The authors go on to propose two
methods that help learn a better bisimulation metric, that result in
representations that more faithfully capture state abstractions from
offline datasets. The two methods are 1) (EBS) an expectile operator
that regularizes the learned state representation to in-sample learning
and 2) (RS) a reward scaling where they tune the hyperparameter on the
reward difference (with theory for further motivation). Experiments are
conducted in offline RL using D4RL with either states or pixel-based
observations. The empirical results provide some evidence that RS+EBS
improve the learned bisimulation metric, and hence the state
representation.

Decision
--------

The theoretical results, at the intersection of bisimulation and offline
RL theory, is interesting and novel. Besides the theory, however, I
remain unconvinced by the empirical evidence and the severity of the
proposed problem. As a result, I lean towards rejection. In the D4RL
experiments, it does not seem that there is a benefit to using the
bisimulation metric at all. While there is a purported benefit in the
visual domain, it is not clear that this is due to RS+EBS or the fact
that there are two additional tunable hyperparameters. Moreover, the two
methods do not really address a novel problem. Missing transitions, and
EBS, is an issue for offline RL in general, not necessarily isolated to the bisimulation metrics. Whereas the
reward scale is an issue with the bisimulation metric, and not necessarily isolated to the offline RL setting.
So, while I like the spirit of the paper, I do not find the empirical evidence
convincing.

**Strengths:**

Strengths
---------

-   Interesting theoretical insights, combining 1) bisimulation theory
    which casts a metric as a fixed point in a lifted MDP and 2) offline
    RL theory which shows limitations of RL algorithms on limited
    datasets.
-   Presentation and initial motivation of the problem, bisimulation in
    the offline setting, is clear. There is value in understanding the
    role of bisimulation metrics, especially in the case of offline RL
    where the learned representation is known to be important for
    transfer.

**Weaknesses:**

Weaknesses
----------

-   While motivation is initially clear, it seems ultimately misguided.
    The results in the reference (\[42\] in paper, Yang and
    Nachum, 2021) does not suggest that bisimulation is particularly
    poor in the offline setting. The bisimulation results are also poor
    in the online setting, and the paper continues to show that several
    methods perform worse in the offline setting, which is unsurprising.
-   The two methods used to improve bisimulation in the offline setting
    are either not unique to issue with bisimulation (missing
    transitions) or not seemingly relevant to the offline setting
    (reward scaling). Both of these methods do seemingly help improve
    learning the bisimulation metric, but the empirical results are not
    convincing that this is helpful over the baselines.
-   Empirical results on D4RL are not entirely convincing, and the
    reasons for excluding MICo+RS are not clear. The results in the
    pixel-based D4RL, while seemingly an improvement on average, are
    difficult to evaluate without confidence intervals and with only 3
    runs.

**Questions:**

Detailed Comments & Questions
-----------------

-   Line 38: \"objectives in most approaches are required to be coupled
    with the policy improvement procedure\" What about their approaches
    require coupling with the policy improvement procedure and why is
    this important? Is this because they are using a $π^*$-bisimulation?

-   Line 39-43: The point you are making here is that bisimulation
    metrics do not work well in offline RL. The points about special
    cases of online RL seem unrelated and are distracting from this main
    point. Moreover, reference \[19\] provides no empirical evidence for
    or against bisimulation metrics in offline RL. Moreover, the
    empirical evidence in \[42\] suggests that all methods do worse in
    the offline RL setting, and does not seem to suggest that
    bisimulation in particular suffers in the offline setting.

-   Line 48: The problem of missing transitions is an inherent problem
    to all offline RL methods.

-   Line 50: The problem of reward scale seems unrelated to offline RL?

-   Lien 55: \"we can achieve a balance between the behavior measurement
    and the greedy assignment of the measurement over the dataset.\" I
    do not understand this statement. Are you claiming that the
    expectile operator balances a type of exploration-exploitation
    trade-off? If so, it is not clear how or why.

-   Line 244: \"Most previous works \[4, 45, 5, 43\] have overlooked the
    impact of reward scaling in the bisimulation\" The DBC paper in
    reference 45 does in fact have a hyperparameter for reward scaling.
    I think this sentence needs more qualification, because most
    combined losses include a hyperparameter for tuning their relative
    importance.

-   Line 248: I have a hard time interpreting the conclusion based off
    of the inequality following (10). The result is an upperbound on the
    fixed point and both $c_k$ and $c_r$ are free variables on the right
    hand of the inequality. But, what does this inequality tells us
    about the trade-off of setting, for example, $c_k = 0$ thus
    upperbounding the fixed point by 0? Different settings of $c_k$ and
    $c_r$ lead to different fixed points, $G^\pi_\tilde$. By using this
    inequality to set an upperbound on the distance, you are also
    biasing the metric. For example, for smaller $c_k$, you are
    considering states similar if the resulting immediate behavior is
    similar and putting less weight on differences in long-term
    behavior.

-   \" However, when bisimulation operators are instantiated with
    bounded distances, e.g., cosine distance, such a setting may be
    unsuitable.\" I don\'t see why this is inherently undesirable. The
    bisimulation distance aggregates the cosine distance across a
    trajectory via the bellman recursion. I do not see why they need to
    have the same upper bound. What you are attempting here seems like,
    in an RL analogy, putting the rewards and the value on the same
    scale.

-   \"with the maximum value of 1 of the cosine distance. To achieve a
    tighter bound in Equation11, we should then maximize the reward
    scale, setting cr to 1 − γ.\" I do not think that c~r~ is a free
    parameter that can be chosen to tighten the bound. This is because
    $c_r$ also determines the fixed point of the bisimulation metric and
    the learned representation. But, accepting your result for the
    moment, I do not see why you set $c_r$ to be $1-\gamma$ and not some
    arbitrarily large number.

-   Analysis of Figure 2: I am not sure what to take away from these
    results. Does the TD3BC baseline use any of the learned
    representations, or is it trained from scratch from the raw state?
    The goal of the paper is understanding and addressing the pitfalls
    of bisimulation representations in offline RL. But, these result
    suggest that bisimulation representations are neither needed nor
    helpful for the downstream task. The motivation is undermined: even
    if bisimulation representations are augmented with your suggested
    changes (expectiles + reward scaling), we get roughly the same
    performance as not using any bisimulation at all.

-   MICo + RS: what does it mean that MICo does not necessitate an
    upper-bound?

-   Table 1: The results are not entirely convincing because the
    pixel-based offline RL experiments are known to have high variance.
    Without reporting confidence intervals, and by uisng only 3 seeds, I
    do not find this evaluation convincing.

-   Ablation: The reward scale coefficient, while motivated by theory,
    is not ablated.

Minor Comments
--------------

-   Line 32: No question has been posed yet, so it is not clear what
    question this paper is answering. A question does occur a few
    paragraphs down, so maybe this statement is an artifact?
-   Figure 2: Some lines are dashed while others are not, and this is
    not indicated in the legend.

**Limitations:**

The authors outline limitations in the expectile operator hyperparameter. Further, I see no potential negative societal concerns.

---

> ### Author Rebuttal · Authors · 2023-08-09
>
> Thank you for your detailed review and feedback on our paper.  We have included new experimental results and some general discussion, please refer to our generic response and uploaded file to address these points.
>
> ## The severity of the proposed problem
>
> Our study focuses on offline state representation learning, instead of offline RL algorithms or offline policy optimization per se. In this paradigm, policy improvement is achieved on top of the learned representation space, and using bisimulation methods to learn representations. Thereby, missing transitions can impact both offline RL algorithms and the bisimulation objective, causing compounding errors. In contrast, other representation objectives [A,B] are not affected by missing transitions since their objective do not explicitly depend on the transitions and underlying dynamics, for which other objectives have so far been shown to be beneficial in offline settings, unlike bisimulation. This is the primary contribution of our work.
>
>  In addition, reward scaling is a crucial concern in the bisimulation principle, where a severe discrepancy may occur between the parameter space of the representation and the policy due to the cumulative effect, intensifying representation collapse in offline environments. In contrast, bisimulation objectives have been shown to be effective for online environments, therefore we take motivation from past works to adapt bisimulation methods to be effective for offline RL.
>
> Additionally, TD3+BC has been shown to be an effective algorithm for rapid convergence and performance in offline RL already; in datasets where TD3+BC struggles to converge, we particularly show the significance of our methodology to be more palpable. Experimental results show that our method converges faster, leading to improved results, on almost 8 out of 12 datasets on standard benchmark settings.
>
> [A]: Mengjiao Yang, Ofir Nachum. Representation Matters: Offline Pretraining for Sequential Decision Making. ICML 2021
>
> [B]: Max Schwarzer, et al. Pretraining Representations for Data-Efficient Reinforcement Learning. NeurIPS 2021
>
> ## Detailed questions
>
> Due to the character limits, we answer per questions below:
>
> - In online settings, most approaches use $\pi$-bisimulation as opposed to $\pi^*$-bisimulation, considering that we cannot ascertain $\pi^*$ during policy improvement. Theoretically, employing bisimulation as a representation objective in online settings mandates iterating bisimulation objectives until they converge to their fixed point at each singular policy $\pi$ during policy iteration. However, this is computationally inefficient. A more comprehensive analysis is available in Appendix A of the DBC paper.
>
> - (Follow above) Conversely, in offline settings, the behavior policy of the offline dataset is generally considered a fixed one. This is the primary distinction between online and offline settings for bisimulation. Consequently, superior performance in online settings should intuitively lead to similar results in offline settings. Paper [42] empirically substantiated the subpar performance of bisimulation in offline settings, and Paper [19] provided a clear corresponding explanation in Section 2.
>
> - We are concerned more with representation learning here instead of offline RL methods themselves. Please refer to the general response.
>
> - In general response.
>
> - No, the expectile operator strikes a balance between behavior and optimality, not exploration-exploitation. We have furnished a background description and insight into utilizing the expectile operator in Appendix C.5.
>
> -  We revise our description to ensure clarity and accuracy. Our objective is to provide theoretical guidance on adjusting reward scaling, rather than introducing a new hyperparameter. While it is true that many combined losses include a hyperparameter to balance their relative importance, our focus is distinct. We aim to guide the balance between immediate similarity and long-term similarity, considering the properties of the associated distance.
>
> - Indeed, a smaller $c_k$ will attribute lesser weight to variations in long-term behavior. When $c_k$ is set to zero, we formulate bisimulation purely via immediate reward, signifying the most greedy way to learn representation. This is precisely why we retain $c_k$ as fixed at $\gamma$, aligning with the settings in value iteration.
>
> - This does not exactly parallel the occurrences for rewards and value. When we aggregate bounded distances such as cosine distance, the precise result we obtain is $cos(\phi(x),\phi(y)) \in [-1,1]$. This indicates that we cannot procure any value outside this boundary, with the upper limit being inaccessible. When considering reward and value, we seldom utilize cosine distance to approximate the value (we scarcely employ tanh activation for value networks either).
>
> - Please refer to the above response and Line 260-270 in our main paper.
>
> - The others please refer to the general response.

---

> > ### Comment · Reviewer_vGKe · 2023-08-18
> >
> > Thanks for the clarifications, I have been convinced at least of the importance of the problem addressed. I accept that missing transitions are also uniquely problematic to Bisimulation-based representations because they are learned via a bellman backup and inherit the flaws of RL methods in offline settings.
> >
> > On the problem of reward scale, however, I remain somewhat puzzled. I do understand that if G is a consine distance, then it may be problematic to learn an embedding that respects G(\phi(x), \phi(y)) \approx |r_x - r_y| + \gamma G(\phi(x'), \phi(y')) (removing expectations for simplicity). This is because  G < 1 while |r_x - r_y| >> 1. But this uses a rather naive implementation of the cosine distance and it seems that the MICo paper adds the norms of the representations in addition to the cosine distance which would prevent this issue (section 5, Castro et al. 2022). Furthermore, adding scaling terms to either the distance or the reward function is not new (original dbc paper had this), and the analysis provided for setting c_r to be 1-\gamma is interesting but ultimately unconvincing. Why should state similarity be weighted differently from value estimation? Put another way, if an agent is trying to optimize the sum of reward for a specific gamma, why should its state representation be using a different weighting?
> >
> > While reward scale can be an issue, it should also be an issue in the online setting and it doesn't seem to be so. At the same time, the results in Figure 2 suggest that reward scaling seems to be a bigger improvement to performance than EBS. Which suggests that the reward scaling chosen is a more pertinent issue than the underlying offline RL problem.
> >
> > Overall, the rebuttal has helped clarify the significance of one of the main problems: that of missing transitions. And there is value in raising this point. However, I still remain unconvinced of the overall submission's contribution and empirical evidence.

---

> > > ### Author Response · Authors · 2023-08-18
> > > **Response to vGKe (1/2)**
> > >
> > > Thank you for your further detailed exploration of our work. Here, we provide point-by-point responses to your questions:
> > >
> > > **Q1: MICo paper adds the norms of the representations in addition to the cosine distance which would prevent this issue (section 5, Castro et al. 2022)...**
> > >
> > > **A1**: This statement is partially true. The standard parameterized distance in MICo is represented by $U_\omega$, which belongs to a partial metric space[1]. This partial metric space yields an associated metric space, where the distance between two points x and y in that metric space is given by $d(x,y) = 2U_\omega(x,y) - U_\omega(x,x) - U_\omega(y,y)$. In practical computation, MICo utilizes the "angular distance"[2] as the associated metric, as described in Section 5 of the MICo paper. It is important to note that MICo introduces a hyperparameter/scalar $\beta$ to scale the angular $\theta$. The choice of $\beta$ can significantly impact the performance, as shown in Figure 13 of the MICo paper. Therefore, instead of relying on the norms of the representations to address this issue, the scalar $\beta$ may help alleviate it, while in the below response, we will show that $\beta$ plays a similar role as reward scaling.
> > >
> > > [1]: Bukatin M, et al. Partial metric spaces[J]. The American Mathematical Monthly, 2009, 116(8): 708-718.
> > >
> > > [2] Wikipedia. Cosine similarity. https://en.wikipedia.org/wiki/Cosine_similarity#Angular_distance_and_similarity.
> > >
> > > **Q2: Adding scaling terms is not new...**
> > >
> > > **A2**: We would like to emphasize for another time that the main contribution of our work is not simply adding a scaling term, but rather providing a theoretical guarantee for its choice. Our contribution can be seen as a theoretical guide that helps researchers understand the relationship between the reward scale and the distance coupled with bisimulation when they opt to use a new distance metric. We acknowledge that previous works have also incorporated such a term, albeit mostly as a user-specified hyperparameter (e.g., DBC and MICo). In offline RL, we generally can't do hyperparameter search so any theoretically motivated heuristic for setting hyperparameters is of high interest.
> > >
> > > **Q3: Why should state similarity be weighted differently from value estimation?**
> > >
> > > **A3**: Firstly, reward rescaling does not have an impact on policy optimization, as supported by Appendix A of DBC's paper. Additionally,  value estimation could use the same reward scaling with no consequence, and therefore there is no inconsistency at this level.
> > >
> > > Secondly, reward scaling aims to align the scale of the reward with the distance used. It can be regarded as a hyperparameter within the representation approach itself. An analogy can be drawn with contrastive learning (CL) methods in computer vision, which also involve multiple hyperparameters, yet their existence does not affect downstream tasks like classification. Similarly, in our case, bisimulation approaches can be seen as analogous to CL, with value function learning and policy improvement being the downstream tasks.
> > >
> > >
> > > **Q4: It should also be an issue in the online setting and it doesn't seem to be so...**
> > >
> > > **A4**: Indeed, this issue also arises in online settings. We additionally constructed an experiment on SimSR in cheetah run tasks ( a DMC task) in online settings. We present some initial results here:
> > >
> > > | Steps    | 10000          | 20000            | 30000           | 40000          | 50000          |
> > > | -------- | -------------- | ---------------- | --------------- | -------------- | -------------- |
> > > | SimSR+RS    | **143.95 (48.71)** | **266.7275 (41.82)**| **394.67 (45.11)** | **446.69 (21.33)** | **557.43 (24.97)** |
> > > | SimSR | 115.81 (30.39) | 237.30 (77.71)   | 362.02  (78.90) | 410.19 (71.36) | 463.09 (27.77) |
> > >
> > > We evaluated the average return over 4 seeds, each column represents the average return at the specific gradient steps. It shows that RS is substantially effective in online settings as well.
> > >
> > > Despite this positive result in the online setting, we would like to stick to our story for the submission. Indeed, our approach is justified by the compounding analysis in the Offline setting. We further support this compounding effect in the Offline setting with an experiment showing that the Bisimulation error may remain high under a small Bellman bisimulation residual error. Finally, we empirically show that RS yields a significant performance improvement in the Offline setting. In our opinion, this forms a well-rounded scientific contribution in itself. We will use this successful but preliminary online experiment to serve as an exciting opening for future studies on reward scaling, both theoretical and empirical. We would like to deeply thank reviewer vGKe for inciting us to investigate the online setting.

---

> > > > ### Author Response · Authors · 2023-08-18
> > > > **Response to vGKe (2/2)**
> > > >
> > > > **Q5: The results in Figure 2 suggest that the reward scaling chosen is a more pertinent issue than the underlying offline RL problem.**
> > > >
> > > > **A5**: The two techniques we have proposed are not parallel but rather cumulative in terms of practical utility. Reward scaling focuses on making bisimulation usable before enhancing its effectiveness; this can be observed in Figure 1 of our recently uploaded PDF file, where the min-max norm enables the usability of SimSR first. Furthermore, the comparison between $c_r=0.001$ and $c_r=0.01$ under the min-max norm demonstrates that it can be further improved to achieve greater effectiveness. Therefore, from this perspective, it is expected that using reward scaling alone would be more effective than using EBS alone in the comparison.
> > > >
> > > > **Q6: I still remain unconvinced of the overall submission's contribution and empirical evidence**
> > > >
> > > > **A6**: Considering the experiments that we designed:
> > > >
> > > > - Figure 2 demonstrated the performance improvement of our method over the baseline bisimulation approaches, highlighting the effectiveness of our techniques and demonstrating the individual effects of each technique as well as their combined effects. Table 1 further illustrates the applicability of our method in scenarios with image observations.
> > > > - Our ablation study, including Figure 5 in Appendix E, supports our theoretical analysis that different $\tau$ affects performance. Additionally, Figure 1 in the newly uploaded file shows that appropriate reward scaling can better match the distance, resulting in improved performance.
> > > >
> > > > Based on these experimental results, we would like to kindly ask if there are any specific aspects we may have overlooked that could have led to a lack of empirical evidence.

---

> > > > > ### Author Response · Authors · 2023-08-20
> > > > > **Kindly Reminder**
> > > > >
> > > > > Dear reviewer,
> > > > >
> > > > > We sincerely appreciate your valuable input and would like to inquire if you have any additional questions or concerns. If there are any, we kindly request that you raise them as soon as possible, allowing us, the authors, sufficient time to address them accordingly.
> > > > >
> > > > > Best,
> > > > >
> > > > > Authors

---

### Author Rebuttal · Authors · 2023-08-09

# General response

## 1.The severity of the proposed problem

### How do bisimulation-based objectives perform in other (online or goal-conditioned) settings?

- Various methods, such as DBC[48], MICo[6], SimSR[46], and PSE[A], have consistently demonstrated positive results in online settings, regardless of the presence of distractors. This evidence supports the efficacy of bisimulation techniques in online settings. Additionally, GCB[B] excelled in goal-conditioned environments, ExTra[C] showcased the power of bisimulation metric in exploration, and HiP-BMDP[D] successfully incorporated bisimulation into multi-task settings, highlighting its superior performance, all mostly in online settings too, with little work in offline RL. These studies suggest that when tailored to specific environments, bisimulation methods can excel. Despite these works, bisimulation methods have had little success when extended to offline settings, and our motivation is to tackle this problem.

### While bisimulation objectives used in the offline setting are directly affected by missing transitions, many other representation objectives may not.

- When referring to state representation learning, using bisimulation in offline settings presents challenges due to the two issues we outlined in Line 48-52 in our original submission: the presence of missing transitions and inappropriate reward scales. Concurrently, there exists other representation objectives, like CURL[28], ATC[E], which focus on pairs of states without the explicit necessity for transition information. As a consequence, they do not explicitly require accounting for missing transitions or reward scaling in their objectives. This absence of direct influence sets them apart from bisimulation-based methods. Yet, we consider that bisimulation-based techniques have a theoretical edge and have proven effective in online settings, Thus, we deem that our work is impactful in that it delivers a proof that bisimulation can be successful offline.

[A]: Rishabh Agarwal, et al. Contrastive Behavioral Similarity Embeddings for Generalization in Reinforcement Learning. ICLR 2021

[B]: Philippe Hansen-Estruch, et al. Bisimulation Makes Analogies in Goal-Conditioned Reinforcement Learning. ICML 2022

[C]: Anirban Santara, et al. ExTra: Transfer-guided Exploration. AAMAS 2020

[D]: Amy Zhang, et al. Multi-Task Reinforcement Learning as a Hidden-Parameter Block MDP. Arxiv 2020.

[E]: Stooke, Adam, et al. Decoupling representation learning from reinforcement learning. ICML 2021.

### Compounded effect for bisimulation principle in offline settings

- In online scenarios, state representations and policies are updated concurrently, while in offline settings, state representation is pre-trained before policy learning, with the two phases completely decoupled. Errors during representation learning in offline settings can have a compounded effect on policy learning, leading to significant issues. This is the reason that missing transitions is particularly harmful to the bisimulation principle in offline settings. Although reward scales affect bisimulation universally, as offline settings require pretraining state embedding, any major discrepancy between this fixed representation and the policy parameter space can further undermine the learning process. Hence, the proposed solutions hold promise for enhancing bisimulation's efficiency in offline settings.


## 2.Suitability of different techniques

- For RS: In essence, the given theoretical analysis is applicable across all bisimulation-based objectives. However, the precise settings for $c_r$ hinge on the foundational distance. For instance, SimSR uses the cosine distance which has definitive bounds. As a result, we need to infer the ideal setting from Equation 10 and Theorem 8. In contrast, the MICo-like distance and DBC employ L-K distance and L1 distance respectively, having bounds ranging from $[0,+\infty]$. Consequently, they can adapt to more value settings. We propose our approach as a general method/principle to employ a novel bisimulation metric or distance, especially in the context of offline RL.

- For EBS: We provide EBS as a general method, which is applicable to all bisimulation-based objectives, given that they all adhere to the foundational principle of bisimulation. This principle revolves around the contraction mapping properties similar to the value iteration. Whenever there's an intent to employ bisimulation in offline scenarios, with an aim to reduce the Bellman residual for approximating the fixed point, the outlined challenge emerges. Consequently, EBS holds the potential to enhance any bisimulation-based method, regardless of the distance they use.

## 3.New experiments results

### MICo+RS and MICo+RS+EBS results

- Please see Figure 2 in rebuttal pdf.  We provide the results of MICo+RS on the D4RL benchmark, as new experiments for the rebuttal.

### Ablations for RS

- In our work, reward scaling comprises two components: i) min-max normalization, and ii) the determination of $c_r$ as $1-\gamma$. To substantiate the efficacy of our proposed methodology, we considered different combinations of min-max normalization/standardization and various value of $c_r$ (including 1, 0.1, 0.01, and 0.001). The results provided empirical validation of our theoretical exploration.

### Visual-D4RL results update
- In the original submission, the methodologies' performance as previously reported in Table 1 omitted variance. We provide additional results in pdf, showing performance curves averaged over 10 different random seeds, and standard error (shared), accompanied by the IQM metric aggregating overall statistical properties. Results in Figures 3 and 4 show consistency with our empirical performance in Table 1. The results depicted in Figure 3 and Figure 4 in the uploaded PDF file show consistency with the empirical performance in Table 1 in our original submission for image-based settings.

---

### Decision · Program_Chairs · 2023-09-21

**Decision:**

Accept (poster)

**Comment:**

### Summary
This paper investigates the limitations of bisimulation based approaches in offline Reinforcement Learning (RL) tasks, where they tend to underperform compared to alternative methods. The authors identify two main challenges: missing transitions in the dataset and reward scaling. Missing transitions can lead to inaccurate estimation of bisimulation metrics and poor state representations. To address this challenge, the paper proposes using the expectile operator for representation learning, which helps prevent overfitting incomplete data. Furthermore, the paper emphasizes the importance of reward scaling in controlling the scale of bisimulation measurements and value errors. To mitigate this issue, the authors introduce a reward scaling strategy. These proposed techniques are applied to two state-of-the-art bisimulation-based algorithms, MICo and SimSR, and evaluated on benchmark suites (D4RL and Visual D4RL), demonstrating performance improvements. Overall, the paper aims to understand why bisimulation methods excel in online settings but struggle in offline RL tasks, offering solutions that enhance the robustness and effectiveness of bisimulation-based representations in offline learning scenarios.

### Decision

The paper is well-written and overall the reviewers were positive about the approach. The paper is well-written and easy to follow. The results on offline RL experiments are quite good. Therefore, I recommend this paper for acceptance.

For the camera-ready I would recommend authors to address the concerns raised by the reviewers with all the additional results presented during the rebuttal.